# Evaluating transportability of in vitro cellular models to in vivo human phenotypes using gene perturbation data

Laurence J. Howe[1] ✉, Yurii S. Aulchenko [1], George Davey Smith [2], Neil M. Davies [3], Jorge Esparza-Gordillo[4], Toby Johnson[1], Jimmy Z. Liu [5], Tom G. Richardson[1], Philippe Sanseau[1], Robert A. Scott [1], Daniel D. Seaton[1], Ashwini Sharma [1] & Adrian Cortes [6]

Gene perturbation screens (e.g. CRISPR-Cas9) assess the impact of gene disruption on in-vitro cellular phenotypes (e.g., proliferation, anti-viral response). In-vitro experiments can be useful models for in-vivo (organismal) phenotypes (e.g., immune cell anti-viral response and infectious diseases). However, assessing whether an in-vitro cellular model effectively captures in-vivo biology is challenging. An in-vitro model is 'transportable' to an in-vivo phenotype if perturbations impacting the in-vitro phenotype also impact the in-vivo phenotype with mechanism-consistent directionality and effect sizes. We propose a framework; Gene Perturbation Analysis for Transportability (GPAT), to assess model transportability using gene perturbation effect estimates from perturbation screens (in-vitro) and loss-of-function burden tests (in-vivo). In hypothesis-driven analyses, GPAT provides evidence for model transportability of higher lysosomal cholesterol accumulation in-vitro to lower human plasma LDL-cholesterol ($P = 0.0006$), consistent with the known role of lysosomes in lipid biosynthesis. In contrast, there was limited evidence for other putative in-vitro models. In hypothesis-free analyses, we find evidence for transportability of cancer cell line proliferation to in-vivo human plasma cellular phenotypes (e.g. erythroleukemia proliferation and plasma lymphocyte percentage). Here we show that perturbation data can be used to evaluate transportability of in-vitro cellular models, informing assay prioritisation and supporting novel hypothesis generation.

In vitro (experimental) cellular models can be useful for understanding the aetiology of in vivo (organismal) phenotypes (e.g., development or progression of a disease). For example, in vitro experiments perturbing gene function and testing for influences on anti-viral responses can serve as a proxy for in vivo anti-viral responses in immune cells, helping to understand the influences on COVID-19 symptoms after SARS-CoV-2 infection[1,2]. The application of gene editing technologies such as CRISPR-Cas9 in functional screens has enabled at-scale investigation of how gene perturbations influence cellular phenotypes[1–10]. If a cellular in vitro model captures pathways involved in disease aetiology, then gene perturbations impacting the in vitro cellular phenotype are putative therapeutic targets for the disease. This has led

[1]GSK, Gunnels Wood Road, Stevenage, United Kingdom. [2]MRC-IEU, University of Bristol, Bristol, UK. [3]Division of Psychiatry, University College London, London, UK. [4]GSK, Calle Severo Ochoa, Tres Cantos, Spain. [5]GSK, Collegeville, Collegeville, Pennsylvania, USA. [6]GSK, Meyerhofstrasse 1, Heidelberg, Germany. ✉e-mail: laurence.x.howe@gsk.com

to a renaissance of interest in phenotypic screening using gene editing technologies. However, the extent to which in vitro cellular models are informative about the aetiology of in vivo phenotypes is often untested and therefore often unclear.

In causal inference, transportability is the extent to which effects estimated in one experimental population correlate with the effects in another population, reflecting stable underlying causal mechanisms[11,12]. Estimates from an in vitro cellular model are 'transportable' to an in vivo outcome phenotype if perturbation effects in the in vitro experiment mirror perturbation effects on the in vivo outcome phenotype. Transportability requires the in vitro model to effectively proxy for in vivo cellular phenotypes, which have downstream effects on the in vivo outcome phenotype (Fig. 1). An in vitro cellular model is an effective proxy if it exhibits highly correlated features with the pathophysiological processes of the in vivo phenotype, e.g., similar biological contexts and cell-type composition. If a model is transportable, we would expect to observe that perturbations impacting the in vitro cellular phenotype also impact the in vivo phenotype with consistent effect sizes and directionality corresponding to the mechanistic relationships between phenotypes.

Genome-wide association study (GWAS) data have been used extensively within causal inference frameworks such as Mendelian randomisation (MR) to evaluate causality between in vivo phenotypes (e.g., LDL-cholesterol measured in plasma and coronary heart disease in humans)[13–15]. The intuition is that if higher LDL-cholesterol increases coronary heart disease risk, then genetic variants associated with higher LDL-cholesterol should also be associated with higher coronary heart disease risk. The MR framework can help to assess transportability between in vitro cellular models and in vivo phenotypes. For example, in an in vitro experimental model where genes in hepatocytes are perturbed, and intracellular lipid content is measured. Assume that the model is transportable for coronary heart disease, i.e.,

(i) the in vitro phenotype effectively proxies for in vivo hepatocytic intracellular lipid content, and (ii) higher in vivo intracellular lipid content increases coronary heart disease risk. It follows that if in vitro perturbation lowering the function of a gene leads to decreased intracellular lipid content, then the same perturbation in vivo should decrease the risk of coronary heart disease. This can be assessed by combining perturbation data with genetic association data in population studies from variants with predicted gene loss-of-function (pLoF) effects.

In vitro experimental screens are generally within a narrow biological context (e.g., specific cellular population and experimental conditions) while germline genetic associations with in vivo human phenotypes capture life-course effects across all cell-types, states and contexts. It follows that germline genetic data can be used to evaluate the general transportability of in vitro models but cannot provide reliable inference relating to specific cellular-level contexts (e.g., is a cell-type disease-relevant) (Fig. 2).

In this work, we outline the Gene Perturbation Analysis for Transportability (GPAT) framework, which uses gene perturbation data to assess model transportability. The framework is generalisable and can be applied to data from different contexts (e.g., in vitro models, mouse models, population studies). In hypothesis-driven analyses, we apply GPAT to evaluate the model transportability of four in vitro models. We find evidence that higher in vitro lysosomal cholesterol accumulation is a transportable model for lower human plasma LDL-cholesterol, but limited evidence for other putative in vitro models (chondrocyte proliferation, adipocyte differentiation and insulin content). In hypothesis-free analyses, we evaluate the transportability of in vitro models from a perturbation screen database with continuous phenotypes in the UK Biobank. We find some evidence for the transportability of cancer cell line proliferation models to in vivo human plasma cellular phenotypes, such as higher

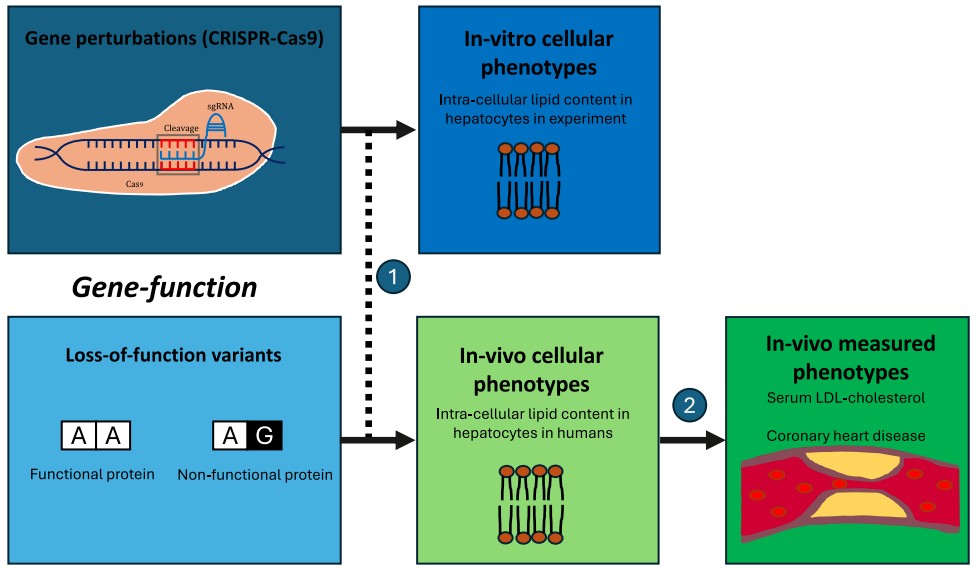

**Fig. 1 | Gene function, in vitro and in vivo phenotypes.** (1) Effects of experimental perturbations on an in vitro cellular phenotype measured under experimental conditions are used as proxies for the effects of genetic LoF variants on a difficult-to-measure in vivo cellular phenotype. In vitro cellular models are generally within a narrow biological context with specific cell types and physiological states. Cellular models may be in non-human model organisms (e.g., mice), which will impact proxy effectiveness. (2) The corresponding in vivo cellular phenotypes can have downstream effects. If we assume that higher in vivo intra-cellular lipid content has a causal effect on increased serum LDL-cholesterol, it follows that genes influencing intra-cellular lipid content also influence serum LDL-cholesterol. For example, if loss-of-function in the LDL receptor gene (*LDLR*) increases intra-cellular lipid

content, then loss-of-function in *LDLR* should also increase serum LDL-cholesterol. In vitro functional screens experimentally perturb gene function to identify genes where reduced (or increased) function affects an in vitro cellular phenotype. In vivo population studies (e.g., in UK Biobank) estimate the effects of germline variants lowering gene function on measured phenotypes in a wide biological context (all cells, across the life-course). GPAT combines these data at the gene level to evaluate the transportability of effect estimates from in vitro cellular models to in vivo measured phenotypes. GPAT estimates capture similarities between the in vitro model and in vivo cellular phenotypes (1) and causal relationships between in vivo phenotypes across a wide biological context (2).

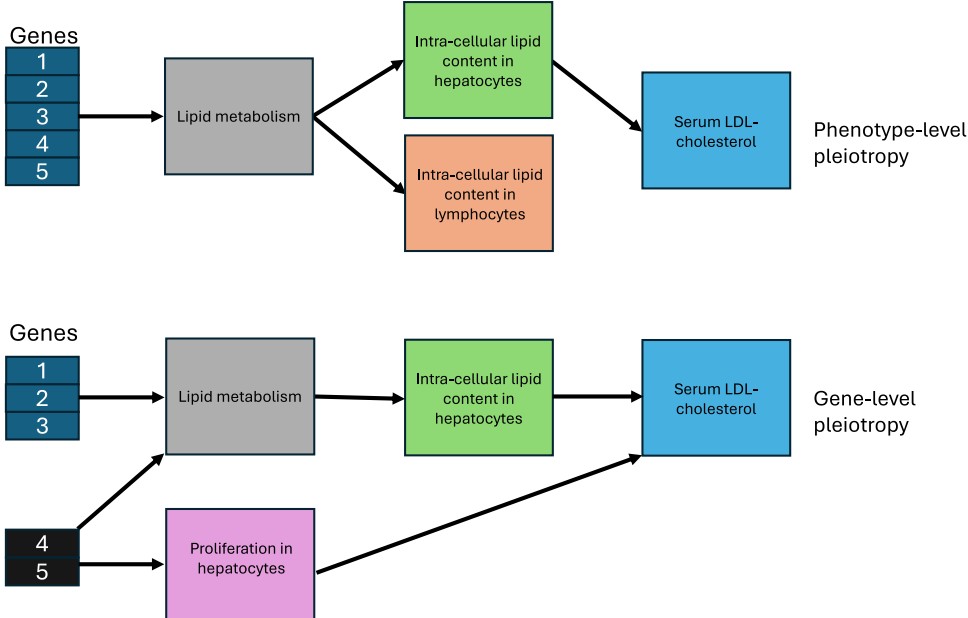

**Fig. 2 | Phenotype-level and gene-level pleiotropy.** In the upper panel of Fig. 2, lipid metabolism influences intra-cellular lipid content in both hepatocytes and lymphocytes. However, only hepatocytes, but not lymphocytes, influence serum LDL-cholesterol. The in vivo associations between genes and serum LDL-cholesterol will capture effects across all biological contexts (e.g., cell-types) and the life-course. Therefore, GPAT estimates can indicate that an in vitro model is transportable but may not necessarily identify the true causal phenotype. We refer to this as 'phenotype-level' pleiotropy (a form of vertical pleiotropy) as it is characterised by alternative paths between genes and the outcome, which are through phenotypes closely related (genetically correlated) to the in vitro model phenotype. In the lower panel of Fig. 2, inter-cellular lipid content in hepatocytes and hepatocyte proliferation have distinct but shared genetic aetiology, with both phenotypes influencing serum LDL-cholesterol. The in vivo associations between genes and serum LDL-cholesterol will capture effects of both phenotypes, and so effects of hepatocyte proliferation may bias GPAT analyses centred on hepatocytic intra-cellular lipid content. However, in this context, the two phenotypes are not perfectly genetically correlated, so we may be able to statistically detect and control for bias relating to this 'gene-level' pleiotropy (a form of horizontal pleiotropy) using methods from the MR literature (e.g., Weighted Median).

erythroleukemia proliferation and lower plasma lymphocyte percentage. Our work illustrates the potential use of perturbation data to evaluate the transportability of in vitro cellular models to human phenotypes.

## Results

### GPAT framework

To illustrate the GPAT framework, suppose one is interested in the transportability of an in vitro cellular model measuring phenotype $k$ to an in vivo phenotype $l$.

(a) In vitro: A functional genomics screen has been conducted, which experimentally perturbed $J$ genes and generated estimates ($\beta_{j \in 1:J, k}$) with corresponding standard errors $\sigma_{j \in 1:J, k}$ of the effect of perturbing gene $j$ on $k$. Experimental knockdown perturbations of a gene generally correspond to a > 80% reduction in gene function.

(b) In vivo: From a population study, we have estimates ($\beta_{j \in 1:J, l}$) with corresponding standard errors $\sigma_{j \in 1:J, l}$ of how germline loss-of-function of $j$ affects $l$. These estimates could come from a rare variant burden test combining predicted pLoF variants into a single score. Heterozygous loss-of-function of a gene (1 unit increase in burden score) is thought to generally correspond to roughly a 50% reduction in gene function.

GPAT is an application of MR using gene perturbations (instead of genetic variants), which are strongly associated in the experimental screen as instrumental variables for the phenotype $k$. Instrumental variables are then used to evaluate transportability between in vitro cellular phenotypes (exposure) and in vivo measured phenotypes (outcome).

If perturbation of the gene $j$ is an instrumental variable, then model transportability can be evaluated using Wald ratios of the estimates from the in vitro and in vivo models, which have coefficients ($\hat{\beta}_j$, $\hat{\sigma}_j^2$):

$$\hat{\beta}_j = \frac{\beta_{j,l}}{\beta_{j,k}} \tag{1}$$

$$\hat{\sigma}_j^2 = \left(\frac{\sigma_{j,l}}{\beta_{j,k}}\right)^2 \tag{2}$$

Wald ratios can then be combined across all instrumental variables using a fixed-effects inverse-variance weighted meta-analysis as follows, where the coefficient $M_\beta$ is the inverse-variance weighted (IVW) estimate of how a 1-unit increase in the in vitro phenotype $k$ corresponds to changes in the in vivo phenotype $l$ with corresponding variance $M_{\sigma^2}$:

$$M_\beta = \frac{\sum_{j \in J}^{J} \frac{\hat{\beta}_j}{\hat{\sigma}_j^2}}{\sum_{j \in J}^{J} \frac{1}{\hat{\sigma}_j^2}} \tag{3}$$

$$M_{\sigma^2} = \frac{1}{\sum_{j \in J}^{J} \frac{1}{\hat{\sigma}_j^2}} \tag{4}$$

Many MR estimators can be applied in the GPAT context (e.g., MR-Egger[16]) and modified versions of the three instrumental variable assumptions of MR apply. First, gene perturbations used as

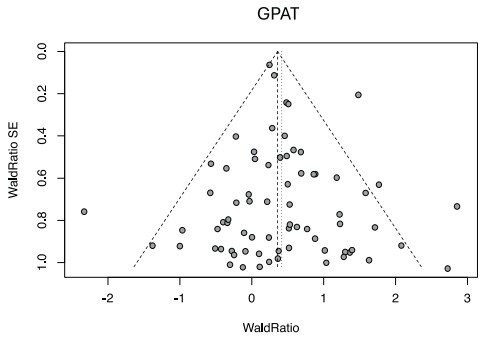
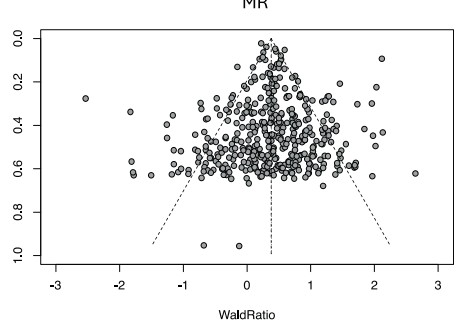

**Fig. 3 | Funnel plot for LDL-cholesterol and coronary heart disease (GPAT and MR).** Wald ratios and standard errors are plotted for gene perturbations (GPAT, left) and genetic variants (MR, right) associated with LDL-cholesterol. Each Wald ratio is an estimate of the effect of LDL-cholesterol on CHD. For example, if pLoF gene burden is associated with a 0.5 unit increase in LDL-cholesterol and a 0.2 log-odds increase in CHD risk, then the Wald ratio for the gene is 0.4. Both GPAT and MR estimates indicate that higher LDL-cholesterol increases the risk of CHD.

The dashed lines represent the fixed-effects (longer bars) and random-effects (shorter bars) meta-analysis estimates and the 95% confidence intervals. For visualisation, we restricted the plot to genes with Wald Ratios less than three and Wald Ratio standard errors less than 1, which contribute the most to the overall estimates. Estimates are presented on the log odds scale in the figure and as odds ratios in the text.

instrumental variables are robustly associated with the in vitro cellular phenotype in the functional screen (relevance). Second, gene perturbations share no common causes with the in vivo outcome phenotype in the population study (independence). For example, ancestry could influence both pLoF variant frequency and the measured phenotypes. Third, gene perturbations affect the in vivo outcome phenotype only via effects of the in vivo cellular exposure, which the in vitro experiment is proxying (exclusion-restriction). In practice, a more tractable assumption is that the net effects of biases on the GPAT estimate are equal to zero (e.g., balanced pleiotropy). Experimental perturbations and pLoF perturbations may not have the same impact on gene function. Therefore, an additional GPAT-specific assumption is that effect estimates from the in vitro and in vivo studies have been rescaled such that the effects reflect equivalent changes in gene function (consistency). Please see the "Methods" for further details, including discussion of GPAT simulation analyses.

### Statistical power of GPAT analyses

The statistical power of GPAT estimates is dependent on several factors: (1) identification of gene perturbations robustly associated with in vitro cellular exposures, (2) variance explained in proxied in vivo cellular exposures by pLoF variants, (3) characteristics of the in vivo outcome phenotype (binary, continuous) and (4) magnitude of effects between in vivo phenotypes.

For (1), characteristics of the experimental screen, such as how many genes were perturbed, perturbation efficiency and the number of guide RNAs used, will impact the discovery of instrumental variables. GPAT with genome-wide screens will have higher statistical power than targeted screens. For (2), pLoF variants can have large effects but are often rare because of negative selection. Therefore, the statistical power of GPAT with genome-wide screen data is likely to be comparable to running MR analyses with low frequency and rare variants. The impact of factors (3) and (4) is well-characterised with binary outcomes having lower power and greater power to detect large effect sizes.

To empirically compare the statistical power of MR and GPAT, we performed positive control analyses using the established in vivo relationship between LDL-cholesterol and coronary heart disease. Using WGS data from UK Biobank, we applied MR and GPAT to evaluate the relationship between serum LDL-cholesterol and coronary heart disease (CHD) (both measured in humans in vivo).

MR using 443 independent genetic variants ($P < 5 \times 10^{-8}$) provided strong evidence that higher LDL-cholesterol increases odds of coronary heart disease, estimating that a 1 SD increase in LDL increases

odds of CHD (OR 1.46; 95% C.I. 1.43, 1.49; $P$-value = $3.4 \times 10^{-244}$). GPAT using 72 genes with pLoF burden evidence (pLoF burden test $P < 0.0005$) provided consistent evidence (OR 1.43; 95% C.I. 1.31, 1.56; $P$-value = $2.1 \times 10^{-16}$) (Fig. 3, Supplementary Table 1). The GPAT estimate standard error was on average ~3.9 times larger than the MR standard error. These results illustrate that GPAT is likely to have lower power than a typical MR analysis but can still have sufficient power for common binary disease phenotypes.

### pLoF burden tests and plasma protein-levels

GPAT assumes that pLoF variants substantially reduce gene function and so are comparable to in vitro gene perturbations. We evaluated the effect of pLoF variants on levels of the corresponding plasma protein for 2121 gene–protein pairs using whole-genome sequence (WGS) pLoF gene burden data from UK Biobank (N ~ 40 K). We found evidence at nominal significance ($P < 0.05$) that pLoF burden on 59.0% (1252) of the tested genes reduces plasma levels of the corresponding protein with large effect sizes (median: 1.3 SD reduction; interquartile range 0.8, 1.8). In contrast, pLoF burden increased plasma protein levels for 42 genes ($P < 0.05$, 1.9%), suggestive of incorrectly inferred directionality (Fig. 4). These results provide evidence that pLoF burden tests can be an effective proxy for reduced gene function, although we were unable to directly compare with the effects of screen perturbations on plasma protein levels. A notable caveat is that inference was based on a subset of protein-coding genes where the encoded protein was measured in the UK Biobank, and which had observed pLoF variants.

**Hypothesis-driven GPAT (in vitro cellular and in vivo human phenotypes).** We next applied GPAT to evaluate the transportability of in vitro cellular models to putatively relevant in vivo human phenotypes using data from previously published genome-wide CRISPR-Cas9 screens. Identification of genes where loss-of-function affects an in vitro cellular phenotype (instruments) requires gene perturbations leading to robust phenotypic changes in the cellular phenotype of interest. Gene perturbation screens can be affected by numerous technical issues, such as off-target effects and variable perturbation efficacy[17,18]. We assumed that the study output was not impacted by or had appropriately controlled for these factors. The impact of gene loss-of-function on in vivo human phenotypes was estimated from pLoF burden tests based on whole genome sequencing of ~500,000 UK Biobank participants ("Methods").

Four experimental screens were selected based on the in vitro phenotype (potential relevance to human in vivo phenotypes) and

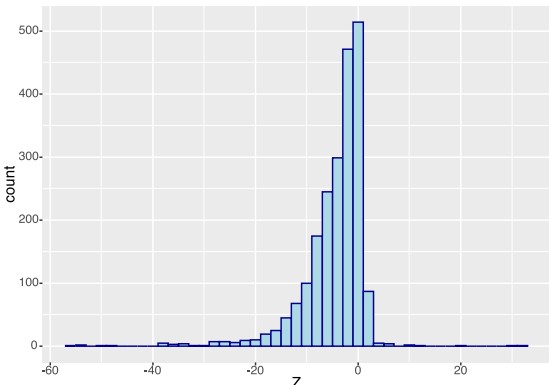
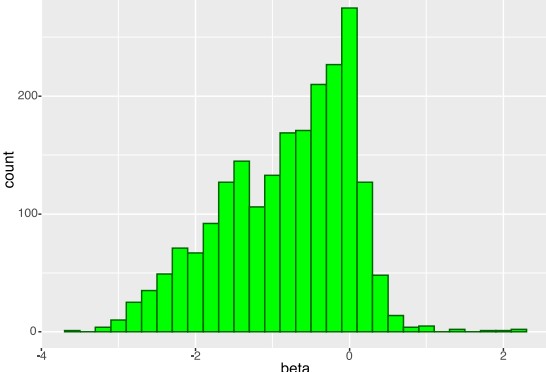

**Fig. 4 | Histogram of distribution of test statistics and effect sizes for associations between pLoF burden of gene and the corresponding protein (plasma levels) pairs.** Histograms of Z scores (left) and signed effect sizes (right) for the association between pLoF burden and encoded protein levels in plasma. The graphs indicate that pLoF lowers plasma-protein levels for most gene-protein pairs, although confidence intervals overlap the null for a substantial proportion, and there are a few outliers in the opposite direction.

### Table 1 | GPAT: effects of in vitro cellular phenotypes on in vivo human phenotypes

| In vitro cellular phenotype | Number of genes[1] | In vivo human phenotype | Enrichment P-value[2] | GPAT estimate[3] (95% C.I.) | GPAT P-value |
|---|---|---|---|---|---|
| Chondrocyte proliferation | 109 | Adulthood height | 0.049 | −0.007 (−0.020, 0.006) | 0.42 |
| Insulin content | 522 | Hba1c | 0.31 | 0.003 (−0.012, 0.019) | 0.48 |
| | | Type II diabetes (OR) | 0.96 | 1.00 (0.99, 1.00) | 0.68 |
| Adipocyte differentiation | 27 | Body mass index | 0.06 | 0.007 (−0.006, 0.021) | 0.30 |
| | | Waist circumference | 0.27 | 0.013 (−0.001, 0.027) | 0.07 |
| | | Body fat percentage | 0.68 | 0.009 (−0.005, 0.023) | 0.20 |
| Lysosomal cholesterol accumulation | 112 | LDL-cholesterol | 0.23 | −0.006 (−0.010, −0.003) | 0.0006 |

[1] Number of instrument genes found in the screen that were matched with genes with pLoF burden test data
[2] Gene-set enrichment P-value for pLoF evidence (see Supplementary Table 2)
[3] Inverse-variance weighted fixed-effects GPAT estimate (two-sided, unadjusted for multiple testing)
Enrichment P-values are based on Chi-squared statistics from two-sided hypothesis tests with no multiple testing adjustment. GPAT P-values are based on Z statistics from two-sided hypothesis tests with no multiple testing adjustment.

data accessibility. Some of the screens are in human cells, and others are in murine cells, which may impact model transportability ("Methods").

1. Chondrocyte proliferation and adulthood height.

   Growth-plate chondrocytes play an aetiological role in bone development, with dysregulated proliferation linked to height-related clinical phenotypes. A previous genome-wide CRISPR screen of chondrocyte proliferation in murine cells demonstrated enrichment of genes involved in height-related phenotypes (e.g., skeletal growth)[8].

2. Intracellular insulin content and type II diabetes (HbA1c).

   Pancreatic beta cells play an important role in the aetiology of type II diabetes. A previous screen evaluated the impact of knockdown on intracellular insulin content in a human pancreatic beta cell line[9].

3. Adipocyte differentiation and adiposity (waist circumference, body mass index, body fat).

   Adipogenesis is the process by which adipocytes accumulate adipose tissue. A previous screen evaluated the impact of gene perturbations on adipocyte differentiation in a murine preadipocyte model[19].

4. Lysosomal cholesterol accumulation and plasma LDL-cholesterol.

   Lysosomes are involved in lipid biosynthesis. A previous screen evaluated the impact of gene perturbations on lysosomal cholesterol accumulation in human K562 cells[20].

Gene-set enrichment analyses were used to evaluate whether in vitro cellular phenotype gene-sets were enriched (relative to other protein-coding genes) for pLoF burden evidence for putatively relevant in vivo human phenotypes. This analysis does not consider directionality or model continuous effect sizes. We found some weak evidence that the chondrocyte proliferation gene-set was positively enriched for genes with pLoF burden evidence for adulthood height (16.5% compared to 10.3%, chi-squared $P = 0.049$). We found limited evidence of enrichment of relevant pLoF-evidence for other cellular gene-sets, but were not sufficiently well-powered to detect modest enrichments (e.g., <two-fold) (Supplementary Table 2).

Next, we performed GPAT analyses to more formally evaluate the transportability of in vitro cellular models to in vivo human phenotypes. We found evidence that higher lysosomal cholesterol accumulation in vitro is a transportable model for lower LDL-cholesterol measured in human blood plasma (beta = −0.006, 95% C.I. −0.010, −0.003; $P = 0.006$) (Table 1, Fig. 5). Estimates from sensitivity analyses were consistent but more imprecisely estimated. There was no strong evidence that estimates were biased by directional pleiotropy ($P = 0.64$) (Supplementary Table 3).

Gene perturbation of *LDLR* in the screen reduced lysosomal cholesterol accumulation by 2.8 units, while loss-of-function of *LDLR* in UK Biobank was associated with an increase in serum LDL-C of 0.79 SD units ($P = 1.7 \times 10^{-40}$). The point estimate ratio for *LDLR* was a visual outlier (annotated in Fig. 5), so we repeated GPAT analyses excluding *LDLR*. The point estimate slightly attenuated but still provided

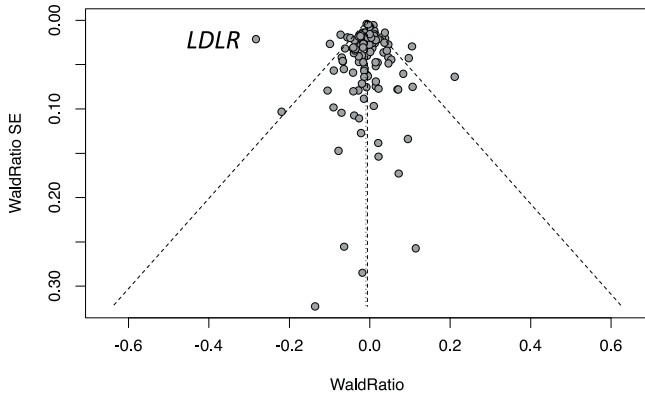

**Fig. 5 | GPAT funnel plot for lysosomal cholesterol accumulation and LDL-cholesterol.** Each data point is the Wald ratio for a gene associated with lysosomal cholesterol accumulation, a point estimate indicating the relationship between lysosomal cholesterol and LDL-cholesterol, with the corresponding standard error. The dashed lines represent the fixed-effects (longer bars) and random-effects (shorter bars) meta-analysis estimates and the 95% confidence intervals. Estimates are presented on the linear scale. The *LDLR* gene is annotated.

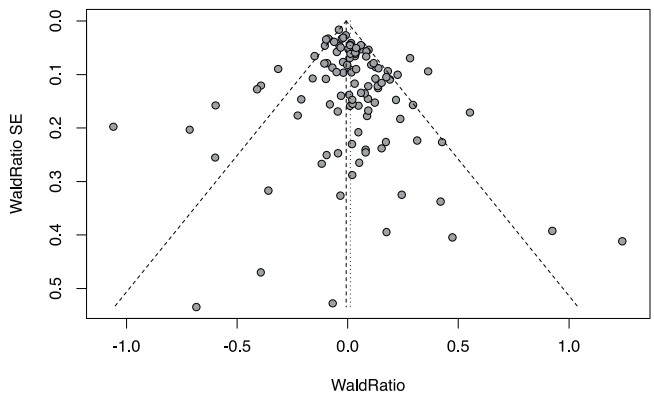

**Fig. 6 | GPAT funnel plot for chondrocyte proliferation genes and height.** Each data point is the Wald ratio for a gene associated with chondrocyte proliferation, a point estimate indicating the relationship between chondrocyte proliferation and height, with the corresponding standard error. The dashed lines represent the fixed-effects (longer bars) and random-effects (shorter bars) meta-analysis estimates and the 95% confidence intervals.

consistent evidence of transportability (beta = −0.004; 95% C.I. −0.008, −0.001; $P = 0.02$).

In contrast, we found limited evidence for the transportability of the other in vitro cellular models to human in vivo phenotypes (Table 1, Supplementary Tables 4–6). For example, we observed some evidence (albeit borderline significant) of enrichment for the chondrocyte-gene set for height pLoF evidence, but found limited evidence when modelling directionality and continuous effect sizes in GPAT that chondrocyte proliferation is a transportable model for height (beta = 0.007 cm; $P$-value = 0.42). Sensitivity analysis estimates were consistent (Supplementary Table 4). Genes where perturbation influenced chondrocyte proliferation were positively enriched for evidence that germline pLoF in humans impacts adulthood height. However, when orienting the directions of effect of the genes on chondrocyte proliferation and height, around half of the genes implied a positive relationship, while the other half implied the opposite, resulting in an overall null estimate in the meta-analysis of Wald ratios. These results suggest that the observed height pLoF evidence enrichment for chondrocyte proliferation genes is likely to reflect gene-level

pleiotropy where common genes influence both chondrocyte proliferation and adulthood height but through distinct mechanisms (Fig. 6).

### Hypothesis-free GPAT (in vitro *cellular and* in vivo *human phenotypes)*

In hypothesis-free analyses, we used experimental screen data from a repository of curated CRISPR screens (BioGRID Open Repository of CRISPR Screens: ORCS). The current release (version 1.1.16, May 2024) includes data from 360 publications, which, after restrictions (Homo Sapiens data, reported MaGECK scores with log2 fold change effect sizes) and quality control, resulted in data for 116 in vitro cellular phenotypes (Supplementary Table 7). Protein/peptide accumulation ($N = 37$), proliferation ($N = 29$) and chemical response ($N = 25$) were the most common in vitro cellular phenotypes, and the majority of screens were in cancer cell lines (> 70 %). GPAT analyses were performed using these in vitro cellular exposures and 69 in vivo continuous outcome phenotypes (e.g., LDL-cholesterol, systolic blood pressure) from UK Biobank ("Methods").

GPAT estimates for 31 in vitro/in vivo phenotype pairs passed a strict Bonferroni multiple-testing correction ($P < 6.2 \times 10^{-6}$) (Supplementary Table 8). The majority ($N = 26$, 84%) of data points involved proliferation in cancer cell lines as the in vitro phenotype. More than half ($N = 20$, 65%) used in vitro exposure data from a single publication, which measured proliferation in 21 different oral cancer cell lines[21]. For 19 of these 20 pairs, GPAT indicated a relationship between higher cancer cell proliferation and lower erythrocyte distribution width in vivo in blood, a relationship supported by evidence from another publication using breast and prostate cancer cells[22].

GPAT indicated transportability between proliferation in an erythroleukemia cell line[23] and in vivo white blood cell phenotypes measured in blood. Higher proliferation was transportable for lower lymphocyte percentage and counts ($P = 1.8 \times 10^{-14}$, $1.8 \times 10^{-5}$) and higher neutrophil percentage and counts ($P = 5.3 \times 10^{-9}$, $1.2 \times 10^{-6}$). Higher proliferation in a gall bladder cancer cell line[24] was also found to be transportable to lower plasma triglycerides ($P = 5.0 \times 10^{-14}$), consistent with the role of the gall bladder in bile storage.

## Discussion

Here we present GPAT a novel framework to evaluate the transportability of in vitro cellular models. Using experimental screen data from the public domain and UK Biobank WGS data, we used a hypothesis-driven approach for a small set of in vitro models and a hypothesis-free approach for a larger set of experimental screens. Improved understanding of the interplay between in vitro and organismal in vivo phenotypes can enable the assessment of assay relevance and predictability and provide opportunities to discover novel targets for the treatment and prevention of disease.

GPAT models both directionality and magnitude of effects, and so is more effective than enrichment analyses for determining if an in vitro model is transportable or if there are simply common genes influencing both phenotypes. GPAT is a gene-level application of MR and so benefits from a suite of existing sensitivity analyses and research. In a positive control analysis, we demonstrated that GPAT using UK Biobank burden test data confirms established relationships between in vivo phenotypes (LDL-cholesterol and coronary heart disease). MR estimates were more precise than GPAT estimates, illustrating that GPAT is likely to have lower power. The main advantage of GPAT is that it can be applied in contexts where experimental perturbations are on the gene-level, whereas MR relies on variant-level estimates.

We found evidence that higher lysosomal cholesterol accumulation in K562 cells is a transportable model for lowering circulating serum LDL-cholesterol, consistent with the role of lysosomes in lipid

biosynthesis[20,25]. GPAT results remained consistent after removing the *LDLR* gene and from other sensitivity analyses. The *LDLR* gene was outlying with the large Wald ratio, suggesting that the *LDLR* gene likely affects serum LDL-cholesterol through additional mechanisms distinct from lysosomal cholesterol accumulation. The in vitro experimental model may only be partially correlated with the true underlying causal in vivo cellular phenotypes, which may feature different cell types or cellular conditions. For growth-plate chondrocyte proliferation and adulthood height, our results indicated that the observed enrichment of genes with LoF height evidence is more likely to be explained by gene-level pleiotropy. We found little evidence of causality for adipocyte differentiation and insulin content with adiposity and type II diabetes-related human phenotypes, respectively. A potential explanation for null findings is that the in vitro cellular phenotype in the functional screen has limited shared aetiology with the in vivo cellular phenotype in humans, indicating low transportability of the in vitro model. For example, height is a complex phenotype that is fixed from early adulthood, and so mirroring in vivo conditions is likely to be challenging. Another explanation is that it can be difficult to measure the most relevant cellular phenotypes. For example, the study authors acknowledged that insulin content may be less disease-relevant than insulin secretion and that the conditions of the screen were under basal glucose conditions[9]. The adiposity and chondrocyte screens used murine models, and so transportability to humans may be lowered by murine-human genetic and phenotypic differences. Further exploration of this cross-species transportability is a necessary component of future work.

In vitro and in vivo settings have some clear distinctions that should be borne in mind. Clearly, whole-organism homoeostatic and allostatic processes[26] will not apply to in vitro cellular models, and this feedback will not be evident in the cellular phenotyping. The time-dependency of in vivo biology is also important to consider. Thus, genetic influences on final height will differ from the foetal growth period, through infancy, childhood and into adulthood. Developmental compensation (canalisation[27]) can lead to insults – whether environmental or due to genetic influences – being compensated for by induced mechanisms (e.g., other genes or pathways). For example, myoglobin plays a key role in cardiac function, but mice with *MB* knockouts exhibit few cardiac abnormalities[28]. This could mean that perturbing a gene experimentally impacts a phenotype, but that germline pLoF is not associated with the phenotype due to such developmental compensation[27]. Time dependency is likely in the chondrocyte proliferation example, where genetic influences on such proliferation in isolated cells at one time point may not reflect the unfolding biology during whole-organism growth and development. These issues again call for further exploration.

The observed null results illustrate the challenges of developing in vitro models for pathophysiological processes. A class of impactful MR studies is those with negative findings consistent with randomised controlled trials, such as limited evidence of a causal effect of HDL-cholesterol on coronary heart disease[29–31]. GPAT can potentially have similar value in demonstrating lack of transportability for an in vitro cellular model, enabling prioritisation of alternative experimental assays.

In hypothesis-free GPAT analyses, we found strong evidence of transportability between proliferation in cancer cell line models and several in vivo human blood cellular phenotypes. Proliferation in cancer cells was found to be transportable to lower erythrocyte width distribution in two different experimental screens. Higher proliferation in an erythroleukemia cell line was transportable to lower blood lymphocytes but higher neutrophils. The interpretation of these results is nuanced, given their cancer-specific context, but they suggest that mechanisms underlying cell proliferation may influence haematopoiesis and characteristics of circulating cells in blood. Most in vitro screen data that we used were in cancer cell lines, and the

phenotypic diversity of the in vivo human phenotypes was also limited. Therefore, our results illustrate only a fraction of the potential of biological inference that is possible if applying GPAT to larger-scale datasets of experimental screens and pLoF burden data.

Unlike MR, GPAT is less focused on effect size estimation, but appropriate inference is still sensitive to the three instrumental variable assumptions (relevance, independence, and exclusion-restriction). Gene perturbations being robustly associated with the cellular phenotype (IV 1: relevance) is unlikely to be a major concern statistically because genome-wide screens are often followed up by validation screens. However, depending on the type 1 error rate control when defining hits in experimental screens, weakly associated gene perturbations can improve power but also bias estimates towards the null (weak-instrument bias). Indeed, our hypothesis-free GPAT analysis used hits defined by study authors and so may be susceptible to weak-instrument bias. Furthermore, for interpretation, it is important to consider the details of the in vitro experiment, such as the source organism (e.g., murine) and state, because this will impact relevance to human in vivo phenotypes. No common causes (IV 2: independence) of loss-of-function variants and human phenotypes is likely a reasonable assumption, with population stratification biasing the pLoF burden tests a possible violation. Violations of the exclusion-restriction assumption (IV 3) include unbalanced horizontal pleiotropy where genes influence the proxied in vivo cellular phenotype and the in vivo human phenotype via different pathways, and this biases the overall estimate. Loss-of-gene function could have many downstream consequences, and so pleiotropy is a concern for GPAT. Indeed, we observed evidence of heterogeneity in most of the GPAT analyses, which could be a consequence of pleiotropy. There are many MR sensitivity analyses that can be used to evaluate unbalanced horizontal pleiotropy to detect and quantify bias. We have applied some of these methods here (Egger, weighted median, weighted mode), finding limited evidence of directional pleiotropy[16,32]. A related caveat is that experimental screens perturb genes in a narrow biological context (e.g., an individual cell-type) while germline pLoF will affect gene function in all cells across the life-course. Therefore, GPAT estimates capture effects of modulation in a wide biological context rather than the specific context and conditions of an in vitro experiment and should not necessarily be extrapolated to specific biological contexts.

A further assumption of GPAT is that gene perturbations (e.g., from CRISPR-Cas9) and pLoF variants present in humans have consistent downstream effects. This is unlikely to hold in practice, as most naturally occurring pLoF carriers are heterozygous, inducing partial loss of function, while gene editing knockouts are likely to have larger (although often variable) effects on gene function. If the relationship between gene function and in vitro / in vivo phenotypes is linear, then this will not impact statistical evidence of causality but can reduce statistical power. Non-linear effects of gene function (e.g., partial loss of function does not affect a phenotype while full loss of function does) could bias GPAT estimates. Using plasma proteomics data, we provided evidence that LoF burden can be an effective instrument for lower gene function, although caveats include possible misannotations and inference being limited to a subset of proteins measured in plasma. Future work could attempt to harmonise the relative effects of LoF variants and gene perturbations on gene function, as these effects are technically measurable.

GPAT is sensitive to problems with input data, such as technical effects in functional screens, such as off-target effects[18] or errors with the prediction of pLoF variants in humans. pLoF variants in highly constrained and/or short in base-pair length genes can be very rare in population biobanks. This will impact statistical power but not necessarily induce bias unless there is a relationship between gene constraint and phenotypic relevance. Unconstrained genes with common germline pLoF variants are likely a non-random subset of all protein-coding genes. This is a potential concern because such genes

will have the largest impact on GPAT estimates due to their more precise pLoF burden estimates. For lower frequency disease phenotypes, the power to detect protective effects is often much lower than to detect adverse effects because of the low frequency of pLoF variants. Therefore, GPAT is likely to be most effective when applied to continuous or common binary human phenotypes.

GPAT enables hypothesis-free and systematic evaluation of the transportability of in vitro cellular models to in vivo organismal phenotypes. The method has limitations and is sensitive to certain assumptions, notably germline human genetic data capture broad organism-level life course effects, which may differ from therapeutic effects in a narrower context. We therefore recommend its application within a triangulation of evidence framework[33] alongside complementary in vitro methods and genomic approaches (e.g., differential expression analysis). The potential applications of GPAT are likely to accelerate over the next decade, driven by increased functional genomics data, larger datasets of sequencing data and improved variant predictions.

## Methods
### GPAT overview
GPAT is an application of MR[15] using gene perturbations (e.g., loss of gene function) as an instrumental variable instead of genetic variants. Many MR concepts (e.g., directional pleiotropy) and estimators (e.g., MR Egger[16]) can be applied with minor differences in interpretation. The three MR instrumental variable assumptions can be adapted for GPAT as follows:

(1) Gene perturbations used as instrumental variables from the experimental screens are robustly associated with the in vitro cellular phenotype in humans (relevance). For example, in MR variants are typically considered as instruments if genome-wide significant ($P < 5 \times 10^{-8}$). The statistics of experimental screens are slightly different from GWAS, but the principles are the same: there should be strong statistical evidence that the gene perturbation is associated with changes in the cellular phenotype (e.g., through a validation screen).

(2) Gene perturbations in the in vivo human genetics study (pLoF variants) share no common causes with the in vivo outcome phenotype (independence). For example, ancestry or cryptic relatedness could influence both LoF variant frequency and in vivo phenotypes. Sufficiently controlling for population stratification and relatedness in the pLoF burden tests should minimise the risk of potential bias.

(3) Gene perturbations influence the in vivo human outcome phenotype only via the proxied in vivo cellular phenotype (exclusion-restriction). Gene-level pleiotropy is ubiquitous, and so in practice this assumption is more that the pleiotropy is 'balanced' in that pleiotropic effects cancel out and so do not bias the overall estimate. Sensitivity analyses like MR Egger can be used to formally test for unbalanced pleiotropy. An additional caveat is that pLoF burden tests will capture effects of cellular phenotypes in all cell types and contexts, while the experimental screen and in vitro exposure are in a narrower biological context. For example, if higher lipid content in hepatocytes lowers plasma LDL-C, then this effect could be masked by higher lipid content increasing plasma LDL-C in other cell-types (if the aetiology of lipid content is highly correlated in both cell-types). The GPAT estimate will capture the net effect of modulating lipid content in all cells rather than specific modulation in hepatocytes. These pleiotropic effects relate to alternative phenotypes that are highly genetically correlated with the phenotype of interest, and so such effects cannot be easily disentangled.

An additional GPAT-specific assumption is that effect estimates from the in vitro and in vivo studies are harmonised such that they reflect equivalent perturbations on gene function (consistency). For example, if the effect of the in vitro experimental perturbation on gene function is 2.5 SD units and the effect of in vivo LoF perturbation on gene function is 1 SD units, then they are scaled accordingly. Perturbation effect estimates from different data sources can be better harmonised using perturb-seq data[34], which measures the impact of perturbations on gene expression. This would enable some rescaling of experimental estimates to a 50% reduction in gene expression for more direct comparison with pLoF estimates. However, such rescaling is sensitive to the assumption of equivalence between gene expression and function and to non-linear effects of changes in function. In practice, perturb-seq data is not always available for all experimental screens, limiting generalisable harmonisation. If perturbation effects are directionally consistent but not perfectly harmonised, then effect estimates can be biased, but causal inference is unlikely to be impacted. If perturbation effects are not directionally consistent (e.g., experimental perturbation increases gene function), then causal inference can be impacted.

If the four assumptions are satisfied, then GPAT can provide unbiased estimates. However, interpretation of GPAT estimates is nuanced and context-specific. In practice, these assumptions are unlikely to hold, and effect sizes are likely to be difficult to directly interpret because of the complexities of measurement and the context of in vitro experiments. We therefore recommend focusing more on directionality and the strength of statistical evidence for GPAT output.

### Simulations
We performed simulations to evaluate the impact of potential biases on GPAT estimates.

In our simulation model, there are:

- 200 genes $G_j$, with each gene having a corresponding unidimensional normally distributed (mean = 0, sd = ) measure of gene function.
- 100 phenotypes of individual cells $X_k$, which are normally distributed (mean = 0, sd = 1). For example, the lipid content of an individual hepatocyte.
- 100 organismal cellular exposure phenotypes $Y_k$, which are normally distributed (mean = 0, sd = 1), $Y_k$ is a function of $X_k$ at the organism-level with one-to-one mapping. For example, the average lipid content across all hepatocytes in an individual human.
- 100 in vivo organismal outcome phenotypes $Z_l$ which are normally distributed (mean = 0, sd = 1). Organism-level phenotypes, such as LDL-cholesterol measured in plasma or coronary heart disease status.

Causal relationships between $G_j$, $X_k$, $Y_k$ and $Z_l$ are simulated as follows:

- $M_C[j,k]$ values are binary values reflecting non-zero or zero effects of $G_j$ on $X_k$ and are sampled from a Binomial distribution ($n = 1$, $p = 0.5$).
- $M_{G \to X}[j,k]$ values are continuous effects of a 1 SD unit increase in $G_j$ on $X_k$ which are conditional on $M_C$ values.

  – If $M_C[j,k] = 0$
    - $M_{G \to X}[j,k] = 0$
  – If $M_C[j,k] = 1$
    - $M_{G \to X}[j,k]$ values are randomly sampled from a Uniform distribution (a = 0.05, b = 0.15) multiplied by a directionality term (1, −1) sampled from a Binomial distribution ($n = 1$, $p = 0.5$).

- Assuming a population of $N$ cells with measured levels of $X_k$, the organism-level measure $Y_k$ is computed as the mean value of $X_k$ across all cells in the cellular population.

$$Y_k = \frac{\sum_{n=1}^{N} X_{k_n}}{N}$$

- $M_{Y \to Z}[k, l]$ values are continuous effects of a 1 SD unit increase in $Y_k$ on $Z_l$ and are simulated as follows.

  - If $k \neq l$
    - $M_{Y \to Z}[k, l] = 0$
  - If $k = l$
    - $M_{Y \to Z}[k, l]$ values are randomly sampled from a Uniform distribution (a = 0.2, b = 0.3).

In vitro experimental perturbation screen data (gene-exposure associations) were simulated as follows:

- All 200 genes are perturbed in the experiment with a vector of perturbation effects $V_j^P$ sampled from a uniform distribution (a = −1.5, b = −1) corresponding to a change in gene function in SD units.
- 10,000 cells are individually perturbed for each gene, with an additional 10,000 cells also from a control experiment with no perturbations.
- $X_{k,j}$ values (values in cells where $G_j$ they have been perturbed) are simulated as follows:

$$X_{k,j} \sim Normal(0, 1) + (M_{G \to X}[j, k] \, V_j^P[j])$$

- $X_{k,C}$ values (values from a control experiment with no perturbations) simulated for comparison:

$$X_{k,C} \sim Normal(0, 1)$$

- $\hat{X}_k$ is the mean value across all individual cells in the in vitro experiment, which is a proxy for the organismal in vivo phenotype $Y_k$.

$$\hat{X}_{k,j} = \frac{\sum_{n=1}^{N} X_{k,j_n}}{N}$$

- $\beta_{j,k}$ the effect of perturbation of a gene $j$ on cellular phenotype $k$ is then estimated using a regression model on the individual cell phenotype values, and is equivalent to the difference in mean values.

$$\beta_{j,k} = \hat{X}_{k,j} - \hat{X}_{k,C}$$

In vivo gene burden data (gene-outcome associations) were simulated as follows:

- All 200 genes have LoF variants.

  100,000 individuals have LoF burden status $\omega_j$ values of 0, 1 or 2 (no LoF variants, one or more heterozygous LoF, one or more homozygous LoF), which are consistent across all cells within the same individual.
  The frequency of LoF burden for each gene $\tau_j$ across the population of individuals is sampled from a Uniform distribution (a = 0.005, b = 0.01).
  The impact of LoF burden on gene function $V_j^{LoF}$ is sampled from a uniform distribution (a = −1, b = −0.5) corresponding to a change in gene function in SD units.

- Each individual $t$ has organism-level cellular exposure $Y_{k,t}$ and outcome $Z_{l,t}$ phenotypes. In the baseline model, we do not simulate $Y_{k,t}$ directly but instead derive the expected deviation in $Y_{k,t}$ of an individual from the population mean. This is based on burden status $\omega_j$, LoF burden impact $V_j^{LoF}$ and the impact of gene function on $X_{k,t}$ ($M_{G \to X}[j, k]$). $Y_{k,t}$ It is defined as the mean value $X_{k,t}$ across all cells, and so the expected change in $Y_{k,t}$ from LoF burden of the gene $j$ is equivalent to $M_{G \to X}[j, k]$.

- We then simulate the outcome $Z_l$ as a product of this expected deviation and the effect of $Y_k$ on $Z_l$ ($M_{Y \to Z}[k, l]$).

$$Z_{l,t} \sim Normal(0, 1) + M_{Y \to Z}[k, l] \left( \sum_{1}^{K} \omega_{j,t} M_{G \to X}[j, k] V_j^{LoF} \right)$$

- $\beta_{j,l}$ (effect of perturbation of gene $j$ on organismal outcome phenotype $l$) values are then estimated using regression models of burden status $\omega_j$ on outcome phenotype values $Z_l$ across all individuals.

- GPAT estimates are generated by:

  Calculating Wald ratios for all genes with corresponding $P$-values using $\beta_{j,k}$ and $\beta_{j,l}$ estimates along with corresponding standard deviations.
  All genes with Wald ratio $P < 0.005$ are included as genetic instruments.
  Perform fixed effects inverse-variance weighted meta-analysis of the genetic instrument Wald ratios for $Y_k$ and $Z_l$.
  As a sensitivity analysis for the detection of unbalanced horizontal pleiotropy, derive the MR Egger estimator estimates.

We first performed GPAT simulation analyses under the baseline model described above.

- Baseline model:

  Apply GPAT to 100 simulated true positive pairs, i.e., the set of $Y_k$-$Z_l$ pairs with causal relationships ($k = l$).
  Gene perturbations $V_j^P$ and $V_j^{LoF}$ have a differential impact on gene function, with LoF burden having a smaller effect on gene function.

- Balanced gene-level pleiotropy model:

  Baseline model with an additional source of variation in $Z_l$ relating to effects of gene burden $\theta[j, k]$ that are not through $Y_k$. In the balanced gene-level pleiotropy model $\theta[j, k]$ is normally distributed with a mean of 0 and a standard deviation of 0.1 and is not associated with $M_{G \to X}[j, k]$.
  Balanced gene-level pleiotropy occurs when the pleiotropic effects of individual genes on the outcome are unlikely to impact GPAT estimates because they are uncorrelated with the effects of genes on the exposure.

$$Z_{l,t} \sim Normal(0, 1) + M_{Y \to Z}[k, l] \left( \sum_{1}^{K} \omega_{j,t} M_{G \to X}[j, k] V_j^{LoF} \right) + \left( \sum_{1}^{K} \omega_{j,t} \theta[j, k] V_j^{LoF} \right)$$

- Unbalanced gene-level pleiotropy model:

  $\theta[j, k]$ is normally distributed with a mean of 0.1 and standard deviation of 0.1, and now multiplied such that its effect direction matches $M_{G \to X}$ (if non-zero). However, the magnitude of $\theta[j, k]$ is not correlated with $M_{G \to X}$ amongst genetic instruments (satisfying the InSIDE assumption)[16].
  Unbalanced gene-level pleiotropy is when the pleiotropic effects of genes on the outcome impact GPAT estimates because they have directional consistency with the effects of the genes on the exposure.

- Phenotype-level pleiotropy model:

  $\theta[j,k]$ is now sampled from a uniform distribution (a = 0.4, b = 0.6) and multiplied by $M_{G \to X}$. Therefore, $\theta[j,k]$ is now positively correlated with $M_{G \to X}$ amongst genetic instruments (violating the InSIDE assumption)[16].
  Phenotype-level pleiotropy is when pleiotropic effects of genes on the outcome impact GPAT estimates because they are positively correlated (in directionality and magnitude) with effects of the genes on the exposure.

- Directional inconsistency model:

  Baseline model with 20% of $V_j^{LoF}$ values having their direction flipped to instead be gain-of-function.
  This results in some experimental perturbations and LoF variants having inconsistent directionalities.

A key element of the baseline model is that differences in perturbation magnitude between experimental (mean = 1.25 SD units) and LoF burden (mean = 0.75 SD units) perturbations were not controlled for. GPAT IVW estimates were biased downwards (expected mean value = 1, observed = 0.56; 95% CI 0.54, 0.58), similar to the ratio of perturbation effects (0.6). The directional inconsistency model showed further attenuations towards the null due to misannotation of pLoF burden estimates for 20% of genes, which actually reflected effects of gain of function (observed mean estimate = 0.36; 95% CI 0.34, 0.38). These results illustrate that GPAT estimates are sensitive to directional consistency and harmonisation in magnitudes between the two perturbation sources (Supplementary Table 9).

Balanced pleiotropy IVW estimates were broadly consistent, while the addition of unbalanced and phenotypic pleiotropic effects led to greatly inflated IVW estimates. These results indicate that GPAT IVW estimates are susceptible to bias from pleiotropic effects if the pleiotropy is non-random with respect to the exposure phenotype.

MR Egger's estimates were more robust against unbalanced pleiotropy with the intercept p-value (median $P = 0.007$) correctly identifying the bias, but as expected, did not detect phenotypic pleiotropy due to violation of the InSIDE assumption. These results indicate that MR sensitivity analyses can be used to detect and control for some forms of pleiotropic bias in GPAT analyses. Please find example GPAT funnel plots for the five simulated models in Supplementary Figs. 6–10.

## Genome-wide perturbation screen data

There is a substantial literature of published gene perturbation screens across a range of in vitro cellular phenotypes. In this manuscript, we performed hypothesis-driven GPAT analyses using data from four studies. These studies were selected based on the following criteria:

a. Putative relevance to human phenotypes measured in the UK Biobank.
b. Genome-wide screens (preferred over targeted screens) to increase the number of genes with data.
c. Accessibility of output data – the level of output data varied substantially across screens in terms of the information provided and the units presented. The heterogeneity of output data made consistent data analysis across different screens challenging.

## Chondrocyte proliferation

Growth-plate chondrocyte proliferation is a putative in vitro model for height. For genetic instruments for chondrocyte proliferation, we extracted a list of 162 genes putatively associated with the cellular phenotype at 4 or 15 days from Table 1 in the original manuscript[8]. We then used the bulk download functions from the ShinyApp browser

(https://chondrocyte.shinyapps.io/Live/) to download results (beta coefficients, p-values) from the primary screen for these genes. The authors noted in the manuscript that the day-4 phenotype was more strongly related to height GWAS than the day-15 phenotype, so we focused on the day-4 screen hits.

The chondrocyte gene-set composed of 126 genes associated with 4-day proliferation in the primary screen (P-value < 0.005) and in the validation screen (P-value < 0.05, consistent directionality). The units of chondrocyte proliferation were $\log_2$ fold changes in single-guide RNA (sgRNA) abundance before and after gene knockdowns. After merging on gene symbol with the human pLoF data, 109/126 (86.5%) of genes had pLoF burden data.

## Insulin content

Insulin content is a putative in vitro model for diabetes-related phenotypes. For genetic instruments for intracellular insulin content, we used 580 genes contained in Supplementary Table 1 of the original manuscript[9]. These genes were selected if sgRNAs demonstrated consistent effects across replicates and if the association met an FDR threshold (< 0.1). The units of intracellular insulin content were based on sgRNA abundance in cells with low insulin content relative to those with high insulin content. The authors did not provide P-values in the supplementary tables, so it was not possible to further refine the provided gene-set based on P-value. After merging on gene symbol with the human pLoF data, 522/580 (90%) of genes had pLoF burden data.

## Adipocyte differentiation

Adipocyte differentiation is a putative in vitro model for adiposity-related phenotypes. We extracted data from Supplementary Table 1 of the manuscript[19], which contained results for 4603 genes and three adipocyte phenotypes (differentiation, lipid accumulation and mitotic clonal expansion). For the purposes of analyses relating to human adiposity phenotypes, we focused on adipocyte differentiation, selecting 32 genes as instruments where perturbation was strongly associated with differentiation ($P < 1 \times 10^{-4}$). We decided on this threshold as a compromise between the strength of the instrument (P-value threshold) and the number of genetic instruments. Experiments were performed in mice, so we used orthology data from the Mouse Genome Database[35] (https://www.informatics.jax.org/homology.shtml) to map mouse genes to corresponding human genes. After merging with the human pLoF data, 27/32 genes (84.4%) had pLoF burden data.

## Lysosomal cholesterol accumulation

Lysosomal cholesterol accumulation is a putative in vitro model for cholesterol and lipid-related phenotypes. We extracted data from Supplementary Table 1 of the manuscript[20], which contained data for 20,525 genes, including annotations, scores and P-values. The table included both "minimum" and "maximum" effect estimates as well as P-values. To reduce possible bias from winner's curse, we used the minimum effect sizes in GPAT and selected 174 genetic instruments with $P < 0.0005$ (threshold compromise between the strength of the instrument and the number of genetic instruments).

When merging with the pLoF data on ensemble gene ID, only 112 genes (64.4%) remained (103 if merging on gene symbol), so we investigated by merging the author gene list with our reference genome (ensemble v106). Merging on ensemble gene ID, we matched 19724/20525 genes (96.1%), of which 18820 (95.4%) were annotated as protein-coding genes. Specifically for the set of 174 genes, 163 (93.7%) were in the reference genome, with 129 (79.1%) annotated as protein-coding, suggesting that 17 protein-coding genes (129−112) did not have sufficient pLoF variants for burden data. The majority of "lost" genes were missing from our reference genome or annotated as non-protein-coding genes.

## ORCS database of Genome-wide perturbation screen data

The BioGRID Open Repository of CRISPR Screens (ORCS) is an open repository of curated CRISPR screens. For hypothesis-free analyses, we downloaded the latest release (version 1.1.16, May 2024), which includes data from 360 publications. To ensure necessary statistical data for GPAT analyses were present, that appropriate statistical analyses had been performed, and to reduce multiple testing burden, we subset the full database as follows:

1. Restricted to Homo Sapiens data.
2. Restricted to studies reporting statistics of interest (MaGECK scores, P-values, $\log_2$ fold change).
3. One or more genes are strongly associated with the cellular phenotype of interest.

This resulted in a subset of 116 in vitro cellular phenotypes from 39 publications (Supplementary Table 7). Protein/peptide accumulation ($N = 37$), proliferation ($N = 29$) and chemical response ($N = 25$) were the most common in vitro cellular phenotypes, and the majority of screens were in cancer cell lines (> 70 %).

## Aggregated rare variant test data in UK Biobank

For GPAT analyses, we require effect estimates of the effect of gene loss-of-function on in vivo human phenotypes. There are a multitude of aggregated rare variant tests (e.g. burden, SKAT, ACAT)[36,37] which use variant masks combining different classes of in-silico predictions (e.g. pLoF, deleterious missense). We decided to use burden tests because they provide signed effect sizes (while most other methods provide only P-values). Any aggregated rare variant test generating signed effect sizes can be used in GPAT. Heterozygous pLoF variants are generally thought to reduce gene function by around 50% while the impact of other deleterious missense variants is less clear. Therefore, we decided to use masks with only high-confidence pLoF variants so that the included variants have as similar effects as possible to in vitro gene perturbations. In principle, the power of GPAT could be improved by including deleterious missense variants, but this may introduce problems relating to the magnitude and directionality of missense effects.

We used pLoF gene burden data from the UK Biobank based on Whole-Genome Sequencing data from 450 K participants of non-Finnish European ancestry[38]. REGENIE (v3.4.1) was used for all analyses[39]. Burden tests were performed using a mask of high-confidence pLoF variants (based on variant-effect predictions: VEP) with MAF < 0.01. The default REGENIE burden model was used, which assigns individuals values of 0 (no heterozygous or homozygous pLoF variants), 1 (one or more heterozygous pLoF variants) or 2 (one or more homozygous pLoF variants).

For quantitative phenotypes, we included 35 blood and urine biochemistry biomarker phenotypes (UK Biobank Showcase Category 17518), 5 anthropometric phenotypes from the baseline assessment data (UK Biobank Showcase Category 100010) and 29 blood cell count phenotypes (UK Biobank Showcase Category 100081) (Supplementary Table 10). Quantitative phenotypes were rank-based inverse-normal transformed prior to analysis.

For disease phenotypes, we included coronary heart disease and type II diabetes. These phenotypes were defined based on a combination of electronic health record data sources from CALIBRE atlas phenotypes[40].

For proteomics, 2923 unique proteins were measured in plasma from 54,219 UKB participants using the antibody-based Olink Explore 3072 PEA. These proteins were then matched to the corresponding gene encoding the protein[41]. In analyses, we used pLoF burden data on 2,121 proteins that had 1 or more pLoF carriers in the study sample.

The UK Biobank has ethical approval from the North West Multi-Centre Research Ethics Committee as a Research Tissue Bank. UK Biobank study participants provided written informed consent for their data to be used in health-related research (https://biobank.ctsu.ox.ac.uk/crystal/crystal/docs/Consent.pdf).

## LDL-CHD MR and GPAT analyses

For the LDL-CHD positive control GPAT analysis, we selected 72 genes as genetic instruments for LDL-C based on pLoF burden test results (P-value < 0.0005 for tests with MAF < 0.01). For the LDL-CHD MR analysis, we selected 443 genetic variants as instruments for LDL-C from a GWAS of LDL-C in UK Biobank (LD clumping: $P < 5 \times 10^{-8}$, $r^2 < 0.001$, MAF < 0.01). For MR and GPAT analyses, we calculated Wald ratios using the equations above and used the R package *metagen* (8.1−0) to generate fixed-effects inverse-variance weighted estimates and funnel plots.

## Enrichment analysis

For enrichment analyses, we created two-by-two tables for the hypothesis-driven analyses. Genes were categorised by in vitro and in vivo evidence for each in vitro / in vivo pair.

(1) Genes where perturbation is strongly associated with the in vitro cellular phenotype (see above for screen-specific details) against all other protein-coding genes.
(2) Genes with in vivo pLoF-evidence (pLoF burden P-value < 0.05) against all other protein-coding genes.

A two-sided chi-squared test was then used to assess gene-set enrichment.

## Hypothesis-driven GPAT analyses

We extracted relevant pLoF burden estimates from the UK Biobank WGS pLoF burden test data for in vitro and in vivo phenotypes described above. As detailed above, some genes, strongly associated with the in vitro cellular phenotype, were lost when merging with the pLoF burden test data. In some instances, the mismatch was because protein-coding genes did not have sufficient pLoF carriers for burden tests, while in other instances, this was because of differences in genome nomenclature or versions or because genes were non-protein-coding.

To reduce bias from misannotation of pLoF variants, we decided to flip (multiplied by −1) the pLoF burden effect-estimate directions for 42 genes where there was evidence (P < 0.05) that pLoF was associated with increased plasma protein levels. This did not have a substantial impact on our estimates as only one of these 42 genes were in the instrument sets for the four cellular phenotypes (*NME3* and intracellular insulin content), and pLoF of this gene was not strongly associated with the relevant in vivo human phenotypes (T2D $P = 0.09$, HbA1c $P = 0.10$). There was very strong statistical evidence that annotated pLoF variants in *NME3* were strongly associated with higher plasma levels of the relevant protein ($\log_{10}$ P-value = 188.7), justifying the correction. We note that we were only able to identify putatively misannotated pLoF variants using the plasma proteomics data for ~10% of protein-coding genes (N ~ 2 K), with a further caveat of low statistical power of pLoF burden tests for some genes.

For GPAT analyses, we calculated Wald ratios using the equations above and used the R package *metagen* to generate fixed-effects and random-effects inverse-variance weighted estimates and funnel plots. We reported the fixed-effects estimates as the primary analysis and the random-effects and heterogeneity estimates in the supplementary material.

For MR Egger, weighted median and weighted mode, we used functions from the *TwoSampleMR* package (0.6.20)[42]. Standard errors for the effect of the experimental perturbations on the in vitro cellular phenotypes are modelled (with a small impact) in these estimators. These data were not available for all cellular phenotypes, so we imputed the standard errors to 0. This will lead to slightly underestimated standard errors for these estimators.

For the lysosomal analysis, we ran an additional leave-one-out sensitivity analysis, rerunning the fixed-effects IVW analyses excluding *LDLR* to evaluate the impact of *LDLR* on the overall estimate.

## Hypothesis-free GPAT analyses

We performed hypothesis-free GPAT analyses using in vitro data from the ORCS database as described above and pLoF burden data from 69 in vivo continuous phenotypes from the UK Biobank. Continuous phenotypes were selected because burden estimates are more precise than for discrete disease phenotypes.

Instruments were selected from in vitro studies based on the original study authors' definitions of a 'hit' (column within the ORCS output) with a further filter on *P*-values < 0.005 to ensure at least some level of association. Here, there is a trade-off between increased power from additional genes and weak-instrument bias towards the null from invalid instrumental variables.

We generated fixed-effects IVW GPAT estimates for the 116 in vitro × 69 in vivo phenotype pairs. For 9 pairs, no estimate was generated because none of the in vitro instruments were found in the corresponding in vivo pLoF burden data. For multiple testing, we used a Bonferroni correction on the number of rows, resulting in a threshold of $6.3 \times 10^{-6}$. This threshold is conservative because many of the in vitro cellular phenotypes are correlated (proliferation in cancer cell lines), and some of the in vivo phenotypes are also highly correlated (lymphocyte counts, lymphocyte percentage). However, our aim was to highlight some of the strongest associations, as context and interpretation are likely to be very experimental, screen-specific, and require detailed deep dives.

## Reporting summary

Further information on research design is available in the Nature Portfolio Reporting Summary linked to this article.

## Data availability

The experimental data used in this study are publicly available. In vitro perturbation data were downloaded from the supplementary material of publications (as described and referenced in the manuscript) and from the BioGRID Open Repository of CRISPR Screens (ORCS) (version 1.1.16, May 2024) website https://orcs.thebiogrid.org/. The human pLoF burden association data used in this study were derived from individual participant data from the UK Biobank. Phenotypic and genotypic data from the UK Biobank are available to researchers via an application https://www.ukbiobank.ac.uk/enable-your-research/apply-for-access. Phenome-wide pLoF burden data (same dataset, similar variant masks) from the UK Biobank WGS dataset[43] are in the AstraZeneca PheWAS browser (https://azphewas.com/).

## Code availability

Standard functions from the TwoSampleMR (0.6.20) and metagen (8.1−0) R packages were used for all empirical GPAT analyses and visualisations as described in the manuscript. Custom R code used for GPAT simulations has been deposited in a public repository (https://zenodo.org/records/17534798). Further queries can be directed to the corresponding author.

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

## Acknowledgements

GDS works within the MRC Integrative Epidemiology Unit at the University of Bristol, which is supported by the Medical Research Council (MC_UU_00032/1).

## Author contributions

L.J.H. conceived the project, performed GPAT analyses and draughted the manuscript. T.G.R. and A.C. performed supporting WGS burden analyses. AS contributed with data extraction for GPAT analyses using ORCS data. L.J.H., Y.S.A., G.D.S., N.M.D., J.E.G., T.J., J.Z.L., T.G.R., P.S., R.A.S., D.D.S., A.S., and A.C. reviewed and critically contributed to the manuscript.

## Competing interests

LJH, YSA, AC, JEG, TJ, JZL, TGR, PS, RAS, DDS, and AS are all employees and/or stockholders of GSK. GDS reports Scientific Advisory Board Membership for Bristol Myers Squibb, Relation Therapeutics and Insitro. The remaining authors declare no competing interests.
