## [Transparent Peer Review file · Nature Communications]

Evaluating transportability of in-vitro cellular models to in-vivo human phenotypes using gene perturbation data

Corresponding Author: Dr Laurence Howe

Version 0:

Reviewer comments:

Reviewer #1

(Remarks to the Author)

This paper presents a new loss-of-function IV framework to evaluate in-vitro cellular phenotypes effects' on organism-level phenotypes. It uses gene perturbation screens data for the exposure and burden test data for the outcome. The proposed framework is conceptually new and interesting, but I have several concerns:

1. The underlying statistical model is unclear to me. Could the authors specify the statistical model as well as the assumptions for their proposed framework? For example, how does the burden test connects the LoF variants and the gene's effect on the organism-level phenotype? Is the gene perturbation screens' effect sizes comparable to the burden test effect size? If not, what is the statistical justification for the proposed framework?
2. Could the authors provide more details on the burden test statistics used in the analysis? There are different types of burden test, and what is the difference if using different burden test in the proposed framework?
3. Related to the statistical foundation for the proposed method, could the authors discuss the impact of invalid IVs as well as in what scenarios invalid IVs may present in their contexts? And is there any potential screening procedure for invalid IVs based on function of genetic variants?
4. Currently no simulation studies to evaluate the performance of proposed method, and the real data analysis seems somewhat weak to me.

Reviewer #2

(Remarks to the Author)

Howe et al. introduce LoF-IV, an interesting framework to assess the relevance of in-vitro cellular phenotypes to organism-level (in-vivo) phenotypes. The framework integrates genome-wide perturbation screen data with loss-of-function (LoF) gene burden data from UK Biobank, and applies Mendelian Randomization (MR) to evaluate whether cellular phenotypes are causally linked to human traits. The study validates the known causal relationship between LDL-cholesterol and coronary heart disease and uncovers a potential link between lysosomal cholesterol accumulation and serum LDL-cholesterol levels, advancing genomics research. I have a few comments detailed below.

1. One notable limitation, as acknowledged by the authors, is the heterogeneity between in-vitro gene perturbations and heterozygous pLoF variants in vivo, which potentially restricts the pLoF burden test to serve as a proxy for screen perturbations. While this study did not directly compare these effects, it would be beneficial to conduct extensive simulation studies to assess whether the MR estimates remain robust under varying degrees of consistency between gene perturbations from functional screens and LoF variants.
2. Given the limited evidence for the relevance of chondrocyte proliferation with adulthood height, intracellular insulin content with type 2 diabetes, and adipocyte differentiation with adiposity, additional analyses incorporating a broader range of cellular and organism-level phenotypes are warranted to improve the generalizability of the proposed framework.
3. I agree that the statistical power of the pLoF burden test may be limited due to the ultra-rare nature of pLoF variants in population biobanks. Have the authors considered incorporating disruptive/deleterious missense variants, or a combination of pLoF and disruptive missense variants (which may also have functional impacts on proteins), as instrumental variables in their framework or data applications? If not, discussing this aspect as a potential extension to the current work could provide valuable insights.

4. Following #3, several methods have been proposed to enhance statistical power compared to the standard burden test, such as the functionally-informed (annotation-weighted) burden test introduced in STAAR [1,2] and the sparse burden association test (SBAT) [3]. The authors are encouraged to discuss these approaches as potential extensions to their proposed framework.

[1] <https://doi.org/10.1038/s41588-020-0676-4>

[2] <https://doi.org/10.1038/s41592-022-01640-x>

[3] <https://doi.org/10.1016/j.ajhg.2024.08.021>

5. In Figure 1, if the authors intend to depict gene perturbations or LoF variants as valid genetic instrumental variables (using a DAG scheme), the arrow connecting gene function to in-vivo phenotypes should be removed to avoid implying a direct causal relationship (horizontal pleiotropy).

6. In the LoF-IV overview section, the authors are encouraged to provide a more detailed explanation of the statistical models underlying the effect size (β_E) and standard error (σ_E) parameters. Additionally, the notation for the effect size and standard error estimates should be updated to $\hat{\beta}_E$ and $\hat{\sigma}_E$ for clarity and consistency.

Reviewer #3

(Remarks to the Author)

The authors propose a novel framework, Loss-of-Function Instrumental Variable Analysis (LoF-IV), to evaluate the relationship between cellular exposures and organism-level phenotypes. This framework integrates estimates of gene effects on cellular phenotypes derived from perturbation screens and estimates of gene effects on organism-level phenotypes obtained from LoF-burden tests. The approach presents an innovative IV method leveraging LoF genes as instruments to investigate causal relationships between exposure traits and outcome diseases. While the idea is novel, there are several methodological concerns and questions that may need further clarification and discussion.

Major Comments and Questions:

1. Consistency of IV-to-exposure effects in the perturbation screening and gene-based GWAS data.

If I understand correctly, the IV-to-exposure statistics are from gene perturbation screens, while the IV-to-outcome statistics are obtained from gene-based GWAS burden tests. A key issue in the use of statistics from two different samples is effect consistency. For standard two-sample MR analyses, an additional assumption is required beyond the core IV assumptions: the IV-to-exposure effects must be consistent across the two samples. So that the IV-to-exposure statistics obtained from the exposure GWAS is literally used as a reference, surrogating for the IV-to-exposure effects in the outcome GWAS data (which is not measured but is assumed to be consistent with the reference). This consistency of IV-to-exposure across two samples is often satisfied in traditional MR analyses when IVs (e.g., genetic variants) are selected with stringent significance criteria, as replication across exposure and outcome GWASs ensures the validity. If not satisfied, the selected IV from the exposure GWAS is not a true instrument. Inference will be invalid.

However, in the context of the proposed LoF-IV framework, LoF genes may have effects in only specific cellular contexts. It might not show consistent IV-to-exposure effects in the gene-to-outcome GWAS data. This potential inconsistency poses a challenge to assess the causal effects from exposure to outcome.

2. How do the authors account for this context-specificity in their analyses?

Have they conducted sensitivity analyses to evaluate the robustness of their results to violations of this assumption?

3. Comparison to existing MR methods using genetic variants as instruments: The goal of IV analysis is to estimate the causal effect of the exposure on the outcome. However, it is not clear what advantages the proposed LoF-IV method offers compared to established MR approaches that use genetic variants as instruments. In standard MR frameworks, thousands of genetic variants with strong effects can serve as instruments, and many of the IV assumptions have been relaxed or adjusted in existing MR methods to improve robustness.

4. What specific scenarios or research questions does the LoF-IV framework address that cannot be effectively tackled by traditional MR approaches? Are there particular biological contexts or data types where the proposed method has a clear advantage?

5. Statistical Power and Limitations of the Burden Test:

The power of IV analysis heavily depends on the strength of the IV-to-outcome association. In the proposed method, the burden test is used to quantify the IV-to-outcome effects. However, the burden test is generally less powerful compared to alternative methods. This may make the LoF-IV method less powerful than genetic variant-based MR approaches.

6. Can the authors conduct a comparative evaluation of the statistical power of their method relative to MR approaches that use genetic variants as IVs?

7. Would it be possible to use SKAT or other gene-based test than burden test?

Other suggestions:

8. Can the authors provide a detailed discussion of the assumptions underlying the LoF-IV framework, particularly focusing on the consistency of IV-to-exposure effects across different samples and cellular contexts.

9. Clarify the specific advantages of the proposed method compared to existing MR approaches using genetic variants as instruments. Highlight the contexts where the LoF-IV framework is most applicable or innovative.

Overall, the manuscript introduces an innovative framework that assesses the causal relationships between cellular and organism-level phenotypes using LoF genes as instruments. While the concept is novel and interesting, the manuscript would benefit from additional methodological clarifications, a more thorough comparison to existing MR methods, and discussions addressing potential limitations in statistical power and assumptions.

Reviewer #4

(Remarks to the Author)

Howe and collaborators propose a framework, loss-of-function instrumental variable analysis (LoF-IV), that aims to evaluate relevance between cellular and organism-level phenotypes by using estimates of gene effects on phenotypes from perturbation screens (cellular) and loss-of-function burden tests (organismal). Methodologically, LoF-IV is in effect a minor modification of common Mendelian randomization (MR) analysis that uses loss of gene function instead of genetic variants as an instrumental (proxy) variable. The authors present results of the application of the method to four selected cellular phenotypes, each paired with one or a few presumably related organismal phenotypes. They found evidence of association only in one case.

The motivation of the authors in this contribution is to present a method to evaluate the relevance of cellular phenotypes to human phenotypes because such a method could help assess the suitability of a given in vitro model to uncover disease-relevant and potentially targetable genetic perturbations. Although I agree that the general problem of investigating the relationship between phenotypic effects of gene perturbations observed in vitro and those occurring in vivo in a human organism is interesting and relevant, in this specific case of a method to estimate relevance, I have concerns about the rationale, clarity of presentation, and the utility and scope of the methodology and presented analyses/results.

Comments below

The framework LoF-IV is proposed throughout the manuscript as a method to evaluate the relevance of cellular phenotypes to human phenotypes. The notion of relevance is ambiguous and context dependent. Therefore, it is not possible to objectively assess whether LoF-IV is an effective general approach to evaluate it. What does it mean for a cellular phenotype to be relevant for human phenotypes?

Both cell cultures and organismal data are human, thus the terminology "human phenotype" is misleading. Cellular versus organismal is more clear.

Throughout the introduction and results, it is not clear at all what LoF-IV is. It is only in the methods section that it becomes clear that LoF-IV is a minor modification to MR analysis. A brief explanation of LoF-IV along with the results, as well as how it relates to the validation tests of enrichment presented as results, would help with clarity.

The authors conclude in the discussion section that "LoF-IV enables hypothesis-free and systematic evaluation of causality between cellular and organism-level phenotypes". If strictly transferring the logic of MR analysis, the aim of LoF-IV would be to infer the causal effect of the cellular phenotype on the organismal phenotype. Is whether or not a cellular phenotype is causally implicated what is considered as relevant? Again this is not clear throughout the manuscript.

I appreciate the discussion of the assumptions and potential limitations of the methodology, and the recommendation of applying it alongside other analyses. I find particularly troublesome the validity of the exclusion restriction assumption. In the present case, this assumption implies that genes that affect a given phenotype in a cellular assay need to affect the organismal phenotype only through that cellular phenotype. In the case of the complex phenotypes analysed in genetic association studies, this assumption is not attainable.

The authors conclude that the observed null results illustrate the challenges of developing disease-relevant in vitro models, and propose that LoF-IV can potentially have value in enabling "greater prioritisation of disease assays" by demonstrating absence of evidence that an in-vitro cellular phenotype is disease-relevant. The consideration of disease-relevance here further complicates matters. Although all diseases are organismal phenotypes, most organismal phenotypes are not diseases. The type of disease considered, its underlying genetic architecture, type of inheritance, how it relates with developmental processes, and when in the life of an individual it clinically manifests will all determine whether cellular assays exist that can be at all informative for the in vivo pathogenic process. All of this complexity might not affect non-disease organismal traits, but how such traits relate to disease is usually not trivial. In fact, of the 7 organismal traits considered by the authors in their analyses (table 1), only one is a disease (T2D). The limited results presented do not support the notion that LoF-IV null results illustrate the challenges of developing disease-relevant in vitro models and can help prioritize disease assays. Authors should consider adding additional analyses to more specifically demonstrate such cases.

Overall, in this reviewer's recommendation that the rationale, limited methodological novelty (slight modification of an existing method), and limited analyses and results presented to demonstrate its utility do not merit publication in Nature Communications. With additional analyses, a specialized journal in the field of genetics could be a better fit.

Version 1:

Reviewer comments:

Reviewer #1

(Remarks to the Author)

The authors have done an excellent job addressing the majority of my comments in this revised manuscript. The responses are generally clear, and the additional explanations and analyses, especially regarding the biological assumptions and methodological details of GPAT, substantially improve the clarity and transparency of the work.

I have two remaining points for further clarification and improvement:

1. In-vitro cellular exposures as a proxy of in-vivo cellular exposures is an assumption rather than something that GPAT can test for? So technically, GPAT can only test (2) in Figure 1A but not (1), is it correct? But "transportability" requires both (1) and (2)?

Relatedly, the authors mentioned several times that in-vitro experiments are usually conducted in a narrow biological context while in-vivo phenotypes are in a wide biological context, and results should be interpreted carefully. Is it correct that GPAT assumes the narrow biological context proximate to the wider context? I am still a bit confused about the interpretation of "transportability" in practice. If an in-vitro model is transportable, what potential biological conclusions can be made? Could the authors clarify it further?

2. The issue of scale mismatch between gene perturbation effects and burden-based LoF effects remains central to understanding how to interpret GPAT results. While the authors now mention a "consistency" assumption and briefly discuss it in the Discussion and Methods, this assumption is crucial to interpreting the meaning of GPAT estimates and should be brought forward earlier in the manuscript (perhaps in the Introduction or early Results).

Is it correct that the baseline model in simulation reflects the scenario where the consistency assumption is violated, and the estimates will be biased, but the inference will be fine?

In particular, it would be valuable to discuss:

a. When the consistency assumption is violated, and what are the consequences? Is the non-linearity of LoF score also a source of violation?

b. The feasibility and future strategies for harmonizing scales between LoF variants and experimental perturbations. Currently, I don't have a good sense of how to harmonize these effects, especially given that the LoF effects might be non-linear.

Reviewer #2

(Remarks to the Author)

Thank you to the authors for thoroughly addressing my previous comments, which has significantly improved the manuscript. I would like to recommend the revised manuscript for publication, with one minor suggestion:

The authors include reference 32 (PMID: 24995866) to summarize existing rare variant tests (e.g., burden, SKAT, ACAT). However, it should be clarified that ACAT is a P-value combination method, while the corresponding rare variant test is ACAT-V, which is not covered in reference 32. For completeness and consistency, I suggest including PMID: 32839606 alongside reference 32, as it summarizes all three tests across variant masks combining different classes of in-silico predictions.

Reviewer #3

(Remarks to the Author)

The authors have insufficiently addressed the concerns raised in my initial review.

1. Causality versus Transportability.

The authors state: "As discussed above, we have moved away from 'causality' terminology to 'transportability' of in-vitro models which negates some of the necessity of strong assumptions."

This response is unsatisfactory. Mendelian Randomization (MR) depends on strong assumptions to infer causality from observational data. Replacing "causality" with "transportability" does not address these fundamental concerns. Instead, it reframes the approach as an association method, which is no longer compatible with an MR framework. If the validity of transportability is uncertain, the experimental perturbations cannot serve as a valid instrument. Instruments, by definition,

must be reliably associated with the exposures of interest. Without this foundation, the proposed method no longer qualifies as a valid MR analysis and would require substantial methodological revision. Accordingly, all words related to MR and instrumental variables should be removed.

2. The newly added simulations are insufficient. They only demonstrate directional consistency, which does not adequately capture the real-world complexity. In practice, heterogeneity between experimental and germline LoF perturbations can lead to effects in opposing directions. These discrepancies pose a serious challenge to the proposed framework and must be more thoroughly investigated.

3. Lack of Comparison with alternatives. The authors claim to have addressed comment #4, but their response is inadequate. Only discussion is not enough. A rigorous comparison with existing methods using empirical data or simulations is essential to demonstrate the advantages of the proposed approach. Without such evidence, the method appears to be just another association-based method, at best an integrative association across data sources, making it difficult to evaluate its validity, robustness, or practical utility relative to current alternatives.

Reviewer #4

(Remarks to the Author)

The authors have made substantial changes to the manuscript; including terminology, explanations, limitations, and new analyses. The manuscript is improved, and I do not have further comments.

Version 2:

Reviewer comments:

Reviewer #1

(Remarks to the Author)

The authors have addressed the concerns raised in my previous round of review. I am satisfied with the paper. Thank you!

Reviewer #3

(Remarks to the Author)

The authors have addressed most of the previous concerns, and the revised manuscript shows substantial improvement in clarity and rigor. The assumptions and limitations are now clearly articulated, and the revised introduction and the added simulations and explanations strengthen the overall presentation. A minor suggestion: The discussion needs to appropriately acknowledge the weaknesses and the sensitivity to assumptions.

REVIEWER COMMENTS

Summary of key changes to the manuscript.

- 1) In an extensive rewrite, we have updated terminology throughout the manuscript, added additional figures and provided further detail on the method and how it compares to Mendelian randomization with additional discussion on strengths and limitations.
 - a. The framework has been renamed from LoF-IV to GPAT (Gene Perturbation Analysis – Transportability) to avoid direct comparisons with instrumental variable analyses which are more focused on effect estimation.
 - b. For causal language, we have removed ‘relevance’ terminology and now define ‘transportability’ (are effect estimates from one sample positively correlated with effects in another sample) as the parameter of interest being estimated by GPAT analyses.

“An *in-vitro* model as ‘transportable’ to an *in-vivo* outcome phenotype if perturbations impacting the *in-vitro* cellular phenotype also impact the *in-vivo* phenotype of interest with consistent effect sizes and directionality corresponding to the mechanistic relationship between *in-vivo* cellular (corresponding to the *in-vitro* cellular phenotype) and *in-vivo* outcome phenotypes. In practice, transportability is when gene perturbation effects on the *in-vitro* phenotype are positively correlated with effects of gene perturbation on the *in-vivo* outcome phenotype. Transportability requires that the *in-vitro* cellular phenotype effectively proxies for *in-vivo* cellular phenotypes. An *in-vitro* cellular model is an effective proxy if it has highly correlated features with pathophysiological processes of the *in-vivo* phenotype, e.g., similar biological contexts in cell-type composition. The second requirement is that modifying the proxied *in-vivo* cellular phenotypes leads to changes in the *in-vivo* outcome phenotype (**Figure 1A**).”

- 2) Additional empirical analyses: hypothesis-free GPAT analyses of 116 *in-vitro* cellular phenotypes and 69 continuous *in-vivo* phenotypes. These analyses provided evidence of transportability of proliferation models in cancer cell lines to *in-vivo* plasma cellular phenotypes.
- 3) GPAT simulations under four different models of pleiotropy to better characterise potential biases.

Reviewer #1 (Remarks to the Author):

This paper presents a new loss-of-function IV framework to evaluate in-vitro cellular phenotypes effects' on organism-level phenotypes. It uses gene perturbation screens data for the exposure and burden test data for the outcome. The proposed framework is conceptually new and interesting, but I have several concerns:

Thank you for the supportive and useful feedback, we have responded below to specific points.

1. The underlying statistical model is unclear to me. Could the authors specify the statistical model as well as the assumptions for their proposed framework? For example, how does the burden test connects the LoF variants and the gene's effect on the organism-level phenotype? Is the gene perturbation screens' effect sizes comparable to the burden test effect size? If not, what is the statistical justification for the proposed framework?

We have made substantial changes to the manuscript to improve the description of the statistical models and how the different model parameters are connected. Notably we have included an overview section on the GPAT framework following the introduction. This section illustrates the premise of GPAT, includes detail on statistical models including equations and discusses how experimental perturbations and pLoF burden impact gene function. Please find an excerpt below.

“To illustrate the GPAT framework, suppose one is interested in the transportability of an *in-vitro* cellular model measuring phenotype k to an *in-vivo* phenotype l .

- (1) *In-vitro*: A functional genomics screen has been conducted which experimentally perturbed J genes and generated estimates $(\beta_{j \in 1:J,k})$ with corresponding standard errors $\sigma_{j \in 1:J,k}$ of the effect of perturbing gene j on k . Experimental knockdown perturbations of a gene generally correspond to a $> 80\%$ reduction in gene function.
- (2) *In-vivo*: From a population study we have estimates $(\beta_{j \in 1:J,l})$ with corresponding standard errors $\sigma_{j \in 1:J,l}$ of how germline loss-of-function of j affects l . These estimates could come from a rare variant burden test combining predicted LoF (pLoF) variants into a single score. Heterozygous LoF of a gene (1 unit increase in burden score) is thought to generally corresponds to roughly a 50% reduction in gene function.”

In the methods section, we have also provided additional data on the specific burden test used.

“We used pLoF gene burden data from UK Biobank based on Whole-Genome Sequencing data from 450K participants of non-Finnish European ancestry³⁰. REGENIE (v3.4.1) was used for all analyses³¹. Burden tests were performed using a mask of high-confidence LoF variants (based on variant-effect predictions: VEP) with MAF < 0.01. The default REGENIE burden model was used which assigns individuals values of 0 (no heterozygous or homozygous pLoF variants), 1 (one or more heterozygous pLoF variants) or 2 (one or more homozygous pLoF variants).”

We have included additional detail on the GPAT assumptions and discussed the differences between MR and GPAT. Please find an excerpt below.

“Many MR estimators can be applied in the GPAT context (e.g., MR-Egger) and modified versions of the three instrumental variable assumptions of MR apply. First, gene perturbations used as instrumental variables are robustly associated with the *in-vitro* cellular phenotype in the functional screen (relevance). Second, gene perturbations share no common causes with the *in-vivo* outcome phenotype in the population study (independence). For example, ancestry could influence both LoF variant frequency and phenotypes. Third, gene perturbations affect the *in-vivo* outcome phenotype only via effects of the *in-vivo* cellular exposure which the *in-vitro* experiment is proxying (exclusion-restriction). In practice, a more tractable assumption is that the net effects of biases on the GPAT estimate are equal to zero (e.g., balanced pleiotropy).”

We have added simulations which investigate the impact of heterogeneity between experimental and germline LoF perturbations on GPAT estimates. The results illustrate that under a model where experimental perturbations have larger effects but are directionally consistent with GPAT perturbations that the statistical inference of GPAT is generally appropriate although effect sizes will be biased downwards. We also note that with additional data in future it can be possible to harmonise effect sizes from gene perturbation screens and burden tests.

“A key element of the baseline model is that differences in perturbation

magnitude between experimental (mean = 1.25 SD units) and LoF burden (mean = 0.75 SD units) perturbations were not controlled for. GPAT IVW estimates were biased downwards (expected mean value = 1, observed = 0.56; 95% CI 0.54, 0.58) similar to the ratio of perturbation effects (0.6). These results illustrate that GPAT estimates are sensitive to directional consistency and harmonization in magnitudes between the two perturbation sources (**Supplementary Table 9**).”

2. Could the authors provide more details on the burden test statistics used in the analysis? There are different types of burden test, and what is the difference if using different burden test in the proposed framework?

By different types of burden tests, we assume the reviewer is referring to aggregated rare variant tests distinct from the classical burden test such as SBAT (sparse burden association test). Our understanding is that any rare variant test producing signed effect sizes can be used in GPAT but that many tests produce only P-values (e.g. SBAT). GPAT requires signed effect sizes so that was the reason for our use of burden tests.

We have added additional detail to the methods section on the exact burden model used for GPAT analyses and justification for these choices (‘Aggregated rare variant test data in UK Biobank’ below).

“For GPAT analyses we require effect estimates of the effect of gene loss-of-function on *in-vivo* human phenotypes. There are a multitude of aggregated rare variant tests (e.g. burden, SKAT, ACAT) ³² which use variant masks combining different classes of in-silico predictions (e.g. LoF, deleterious missense). We decided to use burden tests because they provide signed effect sizes (while most other methods provide only P-values). Any aggregated rare variant test generating signed effect sizes can be used in GPAT. Heterozygous pLoF variants are generally thought to reduce gene function by around 50% while the impact of other deleterious missense variants is less clear. Therefore, we decided to use masks with only high-confidence LoF variants so that included variants have as similar effects as possible to *in-vitro* gene perturbations. In principle, power of GPAT could be improved by including deleterious missense variants, but this may introduce problems relating to magnitude and directionality of missense effects.

We used pLoF gene burden data from UK Biobank based on Whole-Genome Sequencing data from 450K participants of non-Finnish European ancestry³³. REGENIE (v3.4.1) was used for all analyses³⁴. Burden tests were performed using a mask of high-confidence LoF variants (based on variant-effect predictions: VEP) with MAF < 0.01. The default REGENIE burden model was used which assigns individuals values of 0 (no heterozygous or homozygous pLoF variants), 1 (one or more heterozygous pLoF variants) or 2 (one or more homozygous pLoF variants).”

3. Related to the statistical foundation for the proposed method, could the authors discuss the impact of invalid IVs as well as in what scenarios invalid IVs may present in their contexts? And is there any potential screening procedure for invalid IVs based on function of genetic variants?

Inference in GPAT is at the screen-level. For example, there could be two screens using the same reagent, the same cell line and measuring the same phenotype which have different results because of other experimental differences. Therefore, any gene where perturbation is robustly associated with the *in-vitro* phenotype is a valid IV for the screen even if it is because of an experimental bias or off-target effect.

Therefore the primary source of invalid IVs is the statistical approach used to define hits in the screen with respect to type 1 error rate. Theoretically this is relatively lower risk due to the use of secondary and validation screens as well as stringent statistical cut-offs within screens. However, in practice this may occur depending on statistical thresholds used to define hits in experimental screens. Our hypothesis-free GPAT analysis uses hits as defined by the study authors and so may be affected by weak instrument bias. Invalid/weak instruments can lead to estimates being biased towards the null which we have acknowledged in some relevant excerpts below:

Discussion

“Gene perturbations being robustly associated with the cellular phenotype (IV 1: relevance) is unlikely to be a major concern statistically because genome-wide screens are often followed-up by validation screens. However, depending on type 1 error rate control when defining hits in experimental screens, weakly associated gene perturbations can improve power but also bias estimates towards the null (weak-instrument bias). Indeed, our hypothesis-free GPAT analysis used hits defined by study authors and so may be susceptible to weak-instrument bias.”

“GPAT is sensitive to problems with input data such as technical effects in functional screens such as off-target effects¹⁵ or errors with prediction of LoF variants in humans.”

Methods

“Instruments were selected from in-vitro studies based on the original study authors definitions of a ‘hit’ (column within the ORCS output) with a further filter on P-values < 0.005 to ensure at least some level of association. Here there is a trade-off between increased power from additional genes and weak-instrument bias towards the null from invalid instrumental variables.”

4. Currently no simulation studies to evaluate the performance of proposed method, and the real data analysis seems somewhat weak to me.

We have added simulation studies to evaluate GPAT estimators under different pleiotropic models. Please find an excerpt below:

“A key element of the baseline model is that differences in perturbation magnitude between experimental (mean = 1.25 SD units) and LoF burden (mean = 0.75 SD units) perturbations were not controlled for. GPAT IVW estimates were biased downwards (expected mean value = 1, observed = 0.56; 95% CI 0.54, 0.58) similar to the ratio of perturbation effects (0.6). These results illustrate that GPAT estimates are sensitive to directional consistency and harmonization in magnitudes between the two perturbation sources (Supplementary **Table 9**).

Balanced pleiotropy IVW estimates were broadly consistent while the addition of unbalanced and phenotypic pleiotropic effects led to greatly inflated IVW estimates. These results indicate that GPAT IVW estimates are susceptible to bias from pleiotropic effects if the pleiotropy is non-random with respect to the exposure phenotype.

MR Egger estimates were more robust against unbalanced pleiotropy with the intercept p-value (median P = 0.007) correctly identifying the bias but as expected did not detect phenotypic pleiotropy due to violation of the InSIDE assumption. These results indicate that MR sensitivity analyses can be used to detect and control for some forms of pleiotropic bias in GPAT analyses. Please find example GPAT funnel plots for the four simulated models in **Supplementary Figures 6-9.**”

We have also added additional real data analysis performing a hypothesis-free GPAT analysis on 116 *in-vitro* cellular phenotypes from an online data repository of perturbation screens and 69 continuous *in-vivo* phenotypes from UK Biobank. We found some GPAT evidence for transportability of proliferation models in cancer cell lines and *in-vivo* plasma blood phenotypes including some biologically plausible

relationships such as proliferation in erythroleukemia cell lines and plasma lymphocytes. Please find further information in the manuscript: “Hypothesis-free GPAT (in-vitro cellular and in-vivo human phenotypes)”

We hope that these additional analyses have improved the content of the manuscript.

Reviewer #2 (Remarks to the Author):

Howe et al. introduce LoF-IV, an interesting framework to assess the relevance of in-vitro cellular phenotypes to organism-level (in-vivo) phenotypes. The framework integrates genome-wide perturbation screen data with loss-of-function(LoF) gene burden data from UK Biobank, and applies Mendelian Randomization (MR) to evaluate whether cellular phenotypes are causally linked to human traits. The study validates the known causal relationship between LDL-cholesterol and coronary heart disease and uncovers a potential link between lysosomal cholesterol accumulation and serum LDL-cholesterol levels, advancing genomics research. I have a few comments detailed below.

Thank you for the supportive and useful feedback, we have responded below to specific points.

1. One notable limitation, as acknowledged by the authors, is the heterogeneity between in-vitro gene perturbations and heterozygous pLoF variants in vivo, which potentially restricts the pLoF burden test to serve as a proxy for screen perturbations. While this study did not directly compare these effects, it would be beneficial to conduct extensive simulation studies to assess whether the MR estimates remain robust under varying degrees of consistency between gene perturbations from functional screens and LoF variants.

Thank you for this suggestion. We have added GPAT simulations which model the impact of heterogeneity between experimental and germline LoF perturbations on GPAT estimates. The results indicate that if perturbations are directionally consistent but germline LoF perturbations have smaller effects that the statistical inference of GPAT can be generally appropriate but estimates will be biased downwards. We also note that with additional data in future it can be possible to harmonise effect sizes from gene perturbation screens and burden tests.

“A key element of the baseline model is that differences in perturbation magnitude between experimental (mean = 1.25 SD units) and LoF burden (mean = 0.75 SD units) perturbations were not controlled for. GPAT IVW estimates were biased downwards (expected mean value = 1, observed = 0.56; 95% CI 0.54, 0.58) similar to the ratio of perturbation effects (0.6). These results illustrate that GPAT estimates are sensitive to directional consistency and harmonization in magnitudes between the two perturbation sources (**Supplementary Table 9**).”

2. Given the limited evidence for the relevance of chondrocyte proliferation with adulthood height, intracellular insulin content with type 2 diabetes, and adipocyte differentiation with adiposity, additional analyses incorporating a broader range of cellular and organism-level phenotypes are warranted to improve the generalizability of the proposed framework.

We have added additional real data analysis performing a hypothesis-free GPAT analysis on 116 *in-vitro* cellular phenotypes from an online data repository and 69 continuous *in-vivo* phenotypes from UK Biobank. We found some GPAT evidence for transportability of proliferation models in cancer cell lines and *in-vivo* plasma blood phenotypes including some biologically plausible relationships such as proliferation in erythroleukemia cell lines and plasma lymphocytes. Please find further information in the manuscript: “Hypothesis-free GPAT (in-vitro cellular and in-vivo human phenotypes)”

3. I agree that the statistical power of the pLoF burden test may be limited due to the ultra-rare nature of pLoF variants in population biobanks. Have the authors considered incorporating disruptive/deleterious missense variants, or a combination of pLoF and disruptive missense variants (which may also have functional impacts on proteins), as instrumental variables in their framework or data applications? If not, discussing this aspect as a potential extension to the current work could provide valuable insights.

This is an important point, the power of GPAT with pLoF variants only is often modest and could be improved by including additional variants in burden masks.

A key premise of GPAT is that the gene-outcome association (burden test) is comparable to (or can be harmonised with) the gene-exposure (*in-vitro* gene perturbations) estimates. One can be relatively comfortable interpreting heterozygous pLoFs as reducing gene function by ~50% but the impact of deleterious missense variants is less clear. Therefore, we think it is preferable to include only high-confidence pLoF variants which are more likely to have similar effects to *in-vitro* gene perturbations. We could gain additional power by including deleterious missense variants but at the cost of weaker impact of rare variant burden

on gene function. We have added additional detail to the methods section discussing different types of rare variant tests and decisions on variant masks.

“For GPAT analyses we require effect estimates of the effect of gene loss-of-function on *in-vivo* human phenotypes. There are a multitude of aggregated rare variant tests (e.g. burden, SKAT, ACAT) ³² which use variant masks combining different classes of in-silico predictions (e.g. LoF, deleterious missense). We decided to use burden tests because they provide signed effect sizes (while most other methods provide only P-values). Any aggregated rare variant test generating signed effect sizes can be used in GPAT. Heterozygous pLoF variants are generally thought to reduce gene function by around 50% while the impact of other deleterious missense variants is less clear. Therefore, we decided to use masks with only high-confidence LoF variants so that included variants have as similar effects as possible to *in-vitro* gene perturbations. In principle, power of GPAT could be improved by including deleterious missense variants, but this may introduce problems relating to magnitude and directionality of missense effects.

We used pLoF gene burden data from UK Biobank based on Whole-Genome Sequencing data from 450K participants of non-Finnish European ancestry ³³. REGENIE (v3.4.1) was used for all analyses ³⁴. Burden tests were performed using a mask of high-confidence LoF variants (based on variant-effect predictions: VEP) with MAF < 0.01. The default REGENIE burden model was used which assigns individuals values of 0 (no heterozygous or homozygous pLoF variants), 1 (one or more heterozygous pLoF variants) or 2 (one or more homozygous pLoF variants).”

4. Following #3, several methods have been proposed to enhance statistical power compared to the standard burden test, such as the functionally-informed (annotation-weighted) burden test introduced in STAAR [1,2] and the sparse burden association test (SBAT) [3]. The authors are encouraged to discuss these approaches as potential extensions to their proposed framework.

[1] <https://doi.org/10.1038/s41588-020-0676-4>

[2] <https://doi.org/10.1038/s41592-022-01640-x>

[3] <https://doi.org/10.1016/j.ajhg.2024.08.021>

We have clarified in the manuscript that GPAT requires weighted effect sizes so that any aggregated rare variant test which produces these data can be used. However, it is our understanding that STAAR and SBAT produce only p-values and so cannot be used in GPAT.

“For GPAT analyses we require effect estimates of the effect of gene loss-of-function on *in-vivo* human phenotypes. There are a multitude of aggregated rare variant tests (e.g. burden, SKAT, ACAT) ³² which use variant masks combining different classes of in-silico predictions (e.g. LoF, deleterious missense). We decided to use burden tests because they provide signed effect sizes (while most other methods provide only P-values). Any aggregated rare variant test generating signed effect sizes can be used in GPAT.”

5. In Figure 1, if the authors intend to depict gene perturbations or LoF variants as valid genetic instrumental variables (using a DAG scheme), the arrow connecting gene function to in-vivo phenotypes should be removed to avoid implying a direct causal relationship (horizontal pleiotropy).

We have updated Figure 1 (now Figure 1A) as suggested, also making additional changes to clarify that perturbations on the *in-vitro* cellular phenotypes are proxies for perturbation on the same phenotype *in-vivo*.

6. In the LoF-IV overview section, the authors are encouraged to provide a more detailed explanation of the statistical models underlying the effect size (β_E) and standard error (σ_E) parameters. Additionally, the notation for the effect size and standard error estimates should be updated to $\hat{\beta}_E$ and $\hat{\sigma}_E$ for clarity and consistency.

We have added additional equations and improved the terminology through the manuscript. Please find updated equations below from the overview section on the GPAT framework.

If perturbation of gene j is an instrumental variable, then model transportability can be evaluated using Wald ratios which have coefficients ($\hat{\beta}_j, \hat{\sigma}_j^2$):

$$\hat{\beta}_j = \frac{\beta_{j,l}}{\beta_{j,k}}$$

$$\hat{\sigma}_j^2 = \left(\frac{\sigma_{j,l}}{\beta_{j,k}}\right)^2$$

Wald ratios can then be combined across all instrumental variables using a fixed-effects inverse-variance weighted meta-analysis as follows, where the coefficient M_β is the inverse-variance weighted (IVW) estimate of the effect of a 1-unit increase in k on l with corresponding variance M_{σ^2} :

$$M_{\beta} = \frac{\sum_{j \in J} \hat{\beta}_j}{\sum_{j \in J} \frac{1}{\hat{\sigma}_j^2}}$$

$$M_{\sigma^2} = \frac{1}{\sum_{j \in J} \frac{1}{\hat{\sigma}_j^2}}$$

Reviewer #3 (Remarks to the Author):

The authors propose a novel framework, Loss-of-Function Instrumental Variable Analysis (LoF-IV), to evaluate the relationship between cellular exposures and organism-level phenotypes. This framework integrates estimates of gene effects on cellular phenotypes derived from perturbation screens and estimates of gene effects on organism-level phenotypes obtained from LoF-burden tests. The approach presents an innovative IV method leveraging LoF genes as instruments to investigate causal relationships between exposure traits and outcome diseases. While the idea is novel, there are several methodological concerns and questions that may need further clarification and discussion.

Thank you for the supportive and useful feedback, we have responded below to specific points.

Major Comments and Questions:

1. Consistency of IV-to-exposure effects in the perturbation screening and gene-based GWAS data.

If I understand correctly, the IV-to-exposure statistics are from gene perturbation screens, while the IV-to-outcome statistics are obtained from gene-based GWAS burden tests. A key issue in the use of statistics from two different samples is effect consistency. For standard two-sample MR analyses, an additional assumption is required beyond the core IV assumptions: the IV-to-exposure effects must be consistent across the two samples. So that the IV-to-exposure statistics obtained from the exposure GWAS is literally used as a reference, surrogating for the IV-to-exposure effects in the outcome GWAS data (which is not measured but is assumed to be consistent with the reference). This consistency of IV-to-exposure across two samples is often satisfied in traditional MR analyses when IVs (e.g., genetic variants) are selected with stringent significance criteria, as replication across exposure and outcome GWASs ensures the validity. If not satisfied, the selected IV from the exposure GWAS is not a true instrument. Inference will be invalid. However, in the context of the proposed LoF-IV framework, LoF genes may

have effects in only specific cellular contexts. It might not show consistent IV-to-exposure effects in the gene-to-outcome GWAS data. This potential inconsistency poses a challenge to assess the causal effects from exposure to outcome.

2. How do the authors account for this context-specificity in their analyses? Have they conducted sensitivity analyses to evaluate the robustness of their results to violations of this assumption?

This is an important point that experimental screens are perturbing genes in a narrow biological context (specific cell-type, experimental context) while germline pLoF will perturb the gene in a much wider context (all cells across the lifecourse). The gene-outcome estimate models the effect of germline pLoF in all cells on the outcome so GPAT will capture additional unmodelled effects of germline pLoF in other cell-types and contexts (to the experimental screen) on the outcome.

Cellular context is a caveat of using germline genetic evidence in general (rather than specifically this method) to inform tissue-specific therapeutics which activate or deactivate a gene in a narrower biological context. Similarly, a germline pLoF may have been developmentally compensated so is not associated with a phenotype even if experimental perturbation of the same gene does impact the phenotype.

As discussed above, we have moved away from 'causality' terminology to 'transportability' of *in-vitro* models which negates some of the necessity of strong assumptions. GPAT estimates will capture the transportability of screen perturbation effect estimates to germline perturbation effects.

We have rewritten the manuscript to reflect this caveat and provided guidance on appropriate interpretation of GPAT estimates in the introduction (including Figure 1A) discussion and methods:

"In-vitro experimental screens are generally within a narrow biological context (e.g., specific cellular population and experimental conditions) while germline genetic associations with *in-vivo* human phenotypes capture life-course effects across all cell-types, states and contexts. It follows that germline genetic data can be used to evaluate general transportability of *in-vitro* models but cannot provide reliable inference relating to specific cellular-level contexts (e.g. is a cell-type disease-relevant) (**Figure 1B**)."

"A related caveat is that experimental screens perturb genes in a narrow biological context (e.g. an individual cell-type) while germline pLoF will affect gene function in all cells across the life-course. Therefore, GPAT estimates capture effects of modulation in a wide biological context rather than the specific context and conditions of an *in-vitro* experiment and so inferences relating to a narrow biological context should be interpreted with caution."

“An additional caveat is that pLoF burden tests will capture effects of cellular phenotypes in all cell-types and contexts while the experimental screen and *in-vitro* exposure are in a narrower biological context. For example, if higher lipid content in hepatocytes lowers plasma LDL-C then this effect could be masked by higher lipid content increasing plasma LDL-C in other cell-types (if the aetiology of lipid content is highly correlated in both cell-types). The GPAT estimate will capture the net effect of modulating lipid content in all cells rather than specific modulation in hepatocytes. These pleiotropic effects relate to alternative phenotypes that are highly genetically correlated with the phenotype of interest and so such effects cannot be easily disentangled.”

We also added discussion on canalisation, a related caveat:

“An additional concern is developmental compensation (canalization) where impacts on fetal development may be compensated for by feedback mechanisms (e.g. other genes or pathways). For example, myoglobin plays a key role in cardiac function but mice with *MB* knockouts exhibit no cardiac abnormalities²⁷. This could mean that perturbing a gene experimentally impacts a phenotype but that germline pLoFs are not associated with the phenotype²⁸.”

3.Comparison to existing MR methods using genetic variants as instruments: The goal of IV analysis is to estimate the causal effect of the exposure on the outcome. However, it is not clear what advantages the proposed LoF-IV method offers compared to established MR approaches that use genetic variants as instruments. In standard MR frameworks, thousands of genetic variants with strong effects can serve as instruments, and many of the IV assumptions have been relaxed or adjusted in existing MR methods to improve robustness.

4.What specific scenarios or research questions does the LoF-IV framework address that cannot be effectively tackled by traditional MR approaches? Are there particular biological contexts or data types where the proposed method has a clear advantage?

MR and GPAT are similar methods using slightly different data. MR uses data on the genetic-variant level which works well for *in-vivo* phenotypes which can be measured in organisms. Individual genetic variants can be experimentally modified in screens and indeed there are examples. However, with present technologies, this is generally limited to perturbing a small set of variants within a gene (e.g., *BRCA2*) and most genome-wide experimental perturbations are on the gene level. GPAT which is on

the gene / gene perturbation level can be applied to data from these screens allowing insights into the transportability of *in-vitro* models to *in-vivo* phenotypes that would not be possible with MR with current data. Applying gene-level analysis does not necessarily have specific methodological advantages over variant-level analysis, it is more that there does not exist variant-level resolution data in many of the *in-vitro* gene perturbation experiments.

We have discussed potential applications of GPAT in the discussion and how this could be combined with other sources of evidence:

“GPAT enables hypothesis-free and systematic evaluation of the transportability of *in-vitro* cellular models to *in-vivo* organismal phenotypes. The method is sensitive to certain assumptions and so we recommend its application in a triangulation of evidence framework³⁰ alongside complementary *in-vitro* methods and genomic approaches (e.g., differential expression analysis). The potential applications of GPAT are likely to accelerate over the next decade with more functional genomics data, larger datasets of sequencing data and better variant predictions.

5. Statistical Power and Limitations of the Burden Test:

The power of IV analysis heavily depends on the strength of the IV-to-outcome association. In the proposed method, the burden test is used to quantify the IV-to-outcome effects. However, the burden test is generally less powerful compared to alternative methods. This may make the LoF-IV method less powerful than genetic variant-based MR approaches.

6. Can the authors conducted a comparative evaluation of the statistical power of their method relative to MR approaches that use genetic variants as IVs?

We have included discussion of the statistical power of GPAT and MR to the manuscript including an empirical comparison for LDL-C and CHD.

“The statistical power of GPAT estimates is dependent on several factors: (1) identification of gene perturbations robustly associated with *in-vitro* cellular exposures, (2) variance explained in proxied *in-vivo* cellular exposures by LoF variants, (3) characteristics of the *in-vivo* outcome phenotype (binary, continuous) and (4) magnitude of effects between *in-vivo* phenotypes.

For (1), characteristics of the experimental screen such as how many genes were perturbed, perturbation efficiency and the number of sgRNAs used will impact discovery of instrumental variables. GPAT with genome-wide screens will have higher statistical power than targeted screens. For (2), LoF variants

can have large effects but are often rare because of negative selection. Therefore, statistical power of GPAT with genome-wide screen data is likely to be comparable to running MR analyses with low frequency and rare variants. The impact of factors (3) and (4) are well-characterised with binary outcomes having lower power and greater power to detect large effect sizes.

To empirically compare the statistical power of MR and GPAT, we performed positive control analyses using the established *in-vivo* relationship between LDL-cholesterol and coronary heart disease. Using WGS data from UK Biobank, we applied MR and GPAT to evaluate the relationship between serum LDL-cholesterol and coronary heart disease (CHD). MR using 443 independent genetic variants ($P < 5 \times 10^{-8}$) provided strong evidence that higher LDL-cholesterol increases odds of coronary heart disease, estimating that a 1 SD increase in LDL increases odds of CHD (OR 1.46; 95% C.I. 1.43, 1.49; P -value = 3.4×10^{-244}). GPAT using 72 genes with pLoF burden evidence (pLoF burden test $P < 0.0005$) provided consistent evidence (OR 1.43; 95% C.I. 1.31, 1.56; P -value = 2.1×10^{-16}) (**Figure 2 / Supplementary Table 1**). The GPAT estimate standard error was on average ~3.9 times larger than the MR standard error. These results illustrate that GPAT is likely to have lower power than a typical MR analysis but can still have sufficient power for common binary disease phenotypes.”

7. Would it be possible to use SKAT or other gene-based test than burden test?

In theory, it is possible to use any aggregated rare variant test which has directional effect sizes. However, it is our understanding that many gene-based tests such as SKAT generate only P-values. We have added the following to the methods section.

“For GPAT analyses we require effect estimates of the effect of gene loss-of-function on *in-vivo* human phenotypes. There are a multitude of aggregated rare variant tests (e.g. burden, SKAT, ACAT) ²⁹ which use variant masks combining different classes of in-silico predictions (e.g. LoF, deleterious missense). We decided to use burden tests because they provide signed effect sizes (while most other methods provide only P-values). Any aggregated rare variant test generating signed effect sizes can be used in GPAT.”

Other suggestions:

8. Can the authors provide a detailed discussion of the assumptions underlying

the LoF-IV framework, particularly focusing on the consistency of IV-to-exposure effects across different samples and cellular contexts.

We have expanded the exposition of the GPAT assumptions in the methods section. The cellular contexts we discussed the changes made earlier as it related to a previous comment from the same reviewer. Please see below:

“GPAT is an application of MR ¹³ using gene perturbations (e.g. loss of gene function) as an instrumental variable instead of genetic variants. Many MR concepts (e.g., directional pleiotropy) and estimators (e.g., MR Egger ²⁶) can be applied with minor differences in interpretation. The three MR instrumental variable assumptions can be adapted for GPAT as follows:

- (1) Gene perturbations used as instrumental variables from the experimental screens are robustly associated with the *in-vitro* cellular phenotype in humans (relevance). For example, in MR variants are typically considered as instruments if genome-wide significant ($P < 5 \times 10^{-8}$). The statistics of experimental screens are slightly different to GWAS, but the principles are the same, that there should be strong statistical evidence that the gene perturbation is associated with changes in the cellular phenotype (e.g., through a validation screen).
- (2) Gene perturbations in the *in-vivo* human genetics study (pLoF variants) share no common causes with the *in-vivo* outcome phenotype (independence). For example, ancestry or cryptic relatedness could influence both LoF variant frequency and *in-vivo* phenotypes. Sufficiently controlling for population stratification and relatedness in the LoF burden tests should minimise risk of potential bias.
- (3) Gene perturbations influence the *in-vivo* human outcome phenotype only via the proxied *in-vivo* cellular phenotype (exclusion-restriction). Gene-level pleiotropy is ubiquitous and so in practice this assumption is more that the pleiotropy is ‘balanced’ in that pleiotropic effects cancel out and so do not bias the overall estimate. Sensitivity analyses like MR Egger can be used to formally test for unbalanced pleiotropy.

An additional caveat is that pLoF burden tests will capture effects of cellular phenotypes in all cell-types and contexts while the experimental screen and *in-vitro* exposure are in a narrower biological context. For example, if higher lipid content in hepatocytes lowers plasma LDL-C then this effect could be masked by higher lipid content increasing plasma LDL-C in other cell-types (if the aetiology of lipid content is highly correlated in both cell-types). The GPAT estimate will capture the net effect of modulating lipid content in all cells rather than specific modulation in hepatocytes. These pleiotropic effects relate to alternative phenotypes that are highly genetically correlated with the

phenotype of interest and so such effects cannot be easily disentangled.

An additional GPAT specific assumption is that effect estimates from the *in-vitro* and *in-vivo* studies are harmonised such that they reflect equivalent perturbations on gene function (consistency). For example, if the effect of the *in-vitro* experimental perturbation on gene function is 2.5 SD units and the effect of *in-vivo* LoF perturbation on gene function is 1 SD units that they are scaled accordingly. In theory these perturbations could be harmonised (e.g., by measuring impact of perturbations on gene expression) but currently there is insufficient data to do such rescaling at scale.

If the four assumptions are satisfied, then GPAT can provide unbiased estimates. However, interpretation of GPAT estimates is nuanced and context specific. In practice these assumptions are unlikely to hold and effect sizes are likely to be difficult to directly interpret because of complexities of measurement and context of *in-vitro* experiments. We therefore recommend focusing more on directionality and strength of statistical evidence for GPAT output.”

9. Clarify the specific advantages of the proposed method compared to existing MR approaches using genetic variants as instruments. Highlight the contexts where the LoF-IV framework is most applicable or innovative.

Overall, the manuscript introduces an innovative framework that assesses the causal relationships between cellular and organism-level phenotypes using LoF genes as instruments. While the concept is novel and interesting, the manuscript would benefit from additional methodological clarifications, a more thorough comparison to existing MR methods, and discussions addressing potential limitations in statistical power and assumptions.

Thank you for the supportive feedback. I think our response to comment 4 from the same reviewer covers the comparisons between MR and GPAT and the potential applications of GPAT.

Reviewer #4 (Remarks to the Author):

Howe and collaborators propose a framework, loss-of-function instrumental variable analysis (LoF-IV), that aims to evaluate relevance between cellular and organism-level phenotypes by using estimates of gene effects on phenotypes

from perturbation screens (cellular) and loss-of-function burden tests (organismal). Methodologically, LoF-IV is in effect a minor modification of common Mendelian randomization (MR) analysis that uses loss of gene function instead of genetic variants as an instrumental (proxy) variable. The authors present results of the application of the method to four selected cellular phenotypes, each paired with one or a few presumably related organismal phenotypes. They found evidence of association only in one case.

The motivation of the authors in this contribution is to present a method to evaluate the relevance of cellular phenotypes to human phenotypes because such a method could help assess the suitability of a given *in vitro* model to uncover disease-relevant and potentially targetable genetic perturbations. Although I agree that the general problem of investigating the relationship between phenotypic effects of gene perturbations observed *in vitro* and those occurring *in vivo* in a human organism is interesting and relevant, in this specific case of a method to estimate relevance, I have concerns about the rationale, clarity of presentation, and the utility and scope of the methodology and presented analyses/results.

Comments below

Thank you for the supportive and useful feedback, we have responded below to specific points.

The framework LoF-IV is proposed throughout the manuscript as a method to evaluate the relevance of cellular phenotypes to human phenotypes. The notion of relevance is ambiguous and context dependent. Therefore, it is not possible to objectively assess whether LoF-IV is an effective general approach to evaluate it. What does it mean for a cellular phenotype to be relevant for human phenotypes?

Thank you for this comment. We agree that relevance terminology is ambiguous and so are now using 'transportability' which has a clearer definition in the epidemiological literature. We have rewritten the manuscript to update the terminology. Please find the relevant section from the introduction below.

"An *in-vitro* model as 'transportable' to an *in-vivo* outcome phenotype if perturbations impacting the *in-vitro* cellular phenotype also impact the *in-vivo* phenotype of interest with consistent effect sizes and directionality corresponding to the mechanistic relationship between *in-vivo* cellular (corresponding to the *in-vitro* cellular phenotype) and *in-vivo* outcome phenotypes. In practice, transportability is when gene perturbation effects on the *in-vitro* phenotype are positively correlated with effects of gene perturbation on the *in-vivo* outcome phenotype. Transportability requires that the *in-vitro* cellular phenotype effectively proxies for *in-vivo* cellular

phenotypes. An *in-vitro* cellular model is an effective proxy if it has highly correlated features with pathophysiological processes of the *in-vivo* phenotype, e.g., similar biological contexts in cell-type composition. The second requirement is that modifying the proxied *in-vivo* cellular phenotypes leads to changes in the *in-vivo* outcome phenotype (**Figure 1A**)."

Both cell cultures and organismal data are human, thus the terminology "human phenotype" is misleading. Cellular versus organismal is more clear.

Thank you. We have now used organismal and *in-vivo* throughout to distinguish from *in-vitro* noting that cellular phenotypes can be both *in-vitro* and *in-vivo*.

Throughout the introduction and results, it is not clear at all what LoF-IV is. It is only in the methods section that it becomes clear that LoF-IV is a minor modification to MR analysis. A brief explanation of LoF-IV along with the results, as well as how it relates to the validation tests of enrichment presented as results, would help with clarity.

We have made substantial changes to the manuscript, including a section after the introduction which illustrates GPAT. We hope that the GPAT framework and its potential applications is now clearer in the manuscript. Please find a short excerpt below:

"To illustrate the GPAT framework, suppose one is interested in the transportability of an *in-vitro* cellular model measuring phenotype k to an *in-vivo* phenotype l .

- (1) *In-vitro*: A functional genomics screen has been conducted which experimentally perturbed J genes and generated estimates $(\beta_{j \in 1:J,k})$ with corresponding standard errors $\sigma_{j \in 1:J,k}$ of the effect of perturbing gene j on k . Experimental knockdown perturbations of a gene generally correspond to a $> 80\%$ reduction in gene function.
- (2) *In-vivo*: From a population study we have estimates $(\beta_{j \in 1:J,l})$ with corresponding standard errors $\sigma_{j \in 1:J,l}$ of how germline loss-of-function of j affects l . These estimates could come from a rare variant burden test combining predicted LoF (pLoF) variants into a single score. Heterozygous LoF of a gene (1 unit increase in burden score) is thought to generally corresponds to roughly a 50% reduction in gene function.

GPAT is an application of MR using gene perturbations which are strongly associated in the experimental screen (1) as instrumental variables for k . Instrumental variables are then used to evaluate transportability between *in-vitro* cellular phenotypes (exposure) and *in-vivo* measured phenotypes (outcome)."

Re enrichment analyses, we have noted in the text that compared to GPAT analyses, enrichment analyses are a simple analysis which do not model directionality or magnitude of effects.

“Gene-set enrichment analyses were used to evaluate whether *in-vitro* cellular phenotype gene-sets were enriched (relative to other protein-coding genes) for pLoF burden evidence for putatively relevant *in-vivo* human phenotypes. This analysis does not consider directionality or model continuous effect sizes.”

The authors conclude in the discussion section that “LoF-IV enables hypothesis-free and systematic evaluation of causality between cellular and organism-level phenotypes”. If strictly transferring the logic of MR analysis, the aim of LoF-IV would be to infer the causal effect of the cellular phenotype on the organismal phenotype. Is whether or not a cellular phenotype is causally implicated what is considered as relevant? Again this is not clear throughout the manuscript.

This is a good point which we have attempted to address in response to the reviewer’s earlier comment by modifying causal terminology from ‘relevance’ to ‘transportability. We acknowledge that the language is confusing given MR is a ‘causal analysis’. Screen estimates are experiments outside living organisms and so causal language is potentially misleading. The key aim of *in-vitro* studies is to inform the effect of gene activation or deactivation on human phenotypes. Therefore, model transportability from the *in-vitro* model to the *in-vivo* outcome phenotype is an important parameter. We hope that suggested interpretation is now more effectively articulated in the manuscript. Please see excerpts below clarifying the terminology:

“An *in-vitro* model as ‘transportable’ to an *in-vivo* outcome phenotype if perturbations impacting the *in-vitro* cellular phenotype also impact the *in-vivo* phenotype of interest with consistent effect sizes and directionality corresponding to the mechanistic relationship between *in-vivo* cellular (corresponding to the *in-vitro* cellular phenotype) and *in-vivo* outcome phenotypes. In practice, transportability is when gene perturbation effects on the *in-vitro* phenotype are positively correlated with effects of gene perturbation on the *in-vivo* outcome phenotype.”

“GPAT enables hypothesis-free and systematic evaluation of the transportability of *in-vitro* cellular models to *in-vivo* organismal phenotypes. The method is sensitive to certain assumptions and so we recommend its application in a triangulation of evidence framework³⁰ alongside complementary *in-vitro* methods and genomic approaches (e.g., differential expression analysis). The potential applications of GPAT are likely to accelerate over the next decade with more functional genomics data, larger datasets of sequencing data and better variant predictions.

I appreciate the discussion of the assumptions and potential limitations of the methodology, and the recommendation of applying it alongside other analyses. I find particularly troublesome the validity of the exclusion restriction assumption. In the present case, this assumption implies that genes that affect a given phenotype in a cellular assay need to affect the organismal phenotype only through that cellular phenotype. In the case of the complex phenotypes analysed in genetic association studies, this assumption is not attainable.

Yes, in practice, the exclusion restriction assumption is unlikely to hold given ubiquitous gene-level pleiotropy. In practice a more reasonable assumption is that biases cancel out in the overall estimate which has been referred to as 'balanced pleiotropy' in the MR literature. Certain forms of pleiotropy can be detected and controlled for by various MR estimators (e.g. MR Egger). We have modified the text to clarify this and included additional discussion in the methods section:

“Gene perturbations influence the *in-vivo* human outcome phenotype only via the proxied *in-vivo* cellular phenotype (exclusion-restriction). Gene-level pleiotropy is ubiquitous and so in practice this assumption is more that the pleiotropy is 'balanced' in that pleiotropic effects cancel out and so do not bias the overall estimate. Sensitivity analyses like MR Egger can be used to formally test for unbalanced pleiotropy.

An additional caveat is that pLoF burden tests will capture effects of cellular phenotypes in all cell-types and contexts while the experimental screen and *in-vitro* exposure are in a narrower biological context. For example, if higher lipid content in hepatocytes lowers plasma LDL-C then this effect could be masked by higher lipid content increasing plasma LDL-C in other cell-types (if the aetiology of lipid content is highly correlated in both cell-types). The GPAT estimate will capture the net effect of modulating lipid content in all cells rather than specific modulation in hepatocytes. These pleiotropic effects relate to alternative phenotypes that are highly genetically correlated with the phenotype of interest and so such effects cannot be easily disentangled.”

We have also added simulations which evaluate GPAT (MR) estimators (IVW and Egger) under different models of pleiotropy demonstrating that:

- 1) Balanced 'random' pleiotropy will not massively impact GPAT estimates.
- 2) Unbalanced pleiotropy can be detected by sensitivity analyses such as MR Egger depending on the pleiotropic mechanism.

- 3) Correlated 'phenotype' level pleiotropy will not be detected by sensitivity analyses reflecting a limitation of the data resolution.

“A key element of the baseline model is that differences in perturbation magnitude between experimental (mean = 1.25 SD units) and LoF burden (mean = 0.75 SD units) perturbations were not controlled for. GPAT IVW estimates were biased downwards (expected mean value = 1, observed = 0.56; 95% CI 0.54, 0.58) similar to the ratio of perturbation effects (0.6). These results illustrate that GPAT estimates are sensitive to directional consistency and harmonization in magnitudes between the two perturbation sources (**Supplementary Table 9**).

Balanced pleiotropy IVW estimates were broadly consistent while the addition of unbalanced and phenotypic pleiotropic effects led to greatly inflated IVW estimates. These results indicate that GPAT IVW estimates are susceptible to bias from pleiotropic effects if the pleiotropy is non-random with respect to the exposure phenotype.

MR Egger estimates were more robust against unbalanced pleiotropy with the intercept p-value (median $P = 0.007$) correctly identifying the bias but as expected did not detect phenotypic pleiotropy due to violation of the InSIDE assumption. These results indicate that MR sensitivity analyses can be used to detect and control for some forms of pleiotropic bias in GPAT analyses. Please find example GPAT funnel plots for the four simulated models in **Supplementary Figures 6-9**.”

The authors conclude that the observed null results illustrate the challenges of developing disease-relevant in vitro models, and propose that LoF-IV can potentially have value in enabling “greater prioritisation of disease assays” by demonstrating absence of evidence that an in-vitro cellular phenotype is disease-relevant. The consideration of disease-relevance here further complicates matters. Although all diseases are organismal phenotypes, most organismal phenotypes are not diseases. The type of disease considered, its underlying genetic architecture, type of inheritance, how it relates with developmental processes, and when in the life of an individual it clinically manifests will all determine whether cellular assays exist that can be at all informative for the in vivo pathogenic process. All of this complexity might not affect non-disease organismal traits, but how such traits relate to disease is usually not trivial. In fact, of the 7 organismal traits considered by the authors in their analyses (table 1), only one is a disease (T2D). The limited results presented do not support the notion that LoF-IV null results illustrate the challenges of developing disease-relevant in vitro models and can help

prioritize disease assays. Authors should consider adding additional analyses to more specifically demonstrate such cases.

A primary aim of *in-vitro* models is to understand disease aetiology. However, LoF burden tests have very low statistical power for most disease outcomes and so we have mostly used disease-relevant continuous *in-vivo* outcome phenotypes. We have therefore updated terminology throughout avoiding 'disease relevance' and instead referring to human phenotypes more generally.

We have also added additional real data analysis performing hypothesis-free GPAT analyses on 116 *in-vitro* cellular phenotypes from an online data repository and 69 continuous *in-vivo* phenotypes from UK Biobank. We found some GPAT evidence for transportability of proliferation models in cancer cell lines and *in-vivo* plasma blood phenotypes including some biologically plausible relationships such as proliferation in erythroleukemia cell lines and plasma lymphocytes.

Overall, in this reviewer's recommendation that the rationale, limited methodological novelty (slight modification of an existing method), and limited analyses and results presented to demonstrate its utility do not merit publication in Nature Communications. With additional analyses, a specialized journal in the field of genetics could be a better fit.

We have made substantial changes to the manuscript and added additional empirical analyses (hypothesis-free GPAT analyses) and simulations. We hope that the revised manuscript is of greater interest to a general audience.

REVIEWER COMMENTS

Reviewer #1 (Remarks to the Author):

The authors have done an excellent job addressing the majority of my comments in this revised manuscript. The responses are generally clear, and the additional explanations and analyses, especially regarding the biological assumptions and methodological details of GPAT, substantially improve the clarity and transparency of the work.

Thank you to the reviewer for the useful feedback.

I have two remaining points for further clarification and improvement:

1. In-vitro cellular exposures as a proxy of in-vivo cellular exposures is an assumption rather than something that GPAT can test for? So technically, GPAT can only test (2) in Figure 1A but not (1), is it correct? But "transportability" requires both (1) and (2)?

Previously we did not provide sufficient detail on transportability. We have rewritten the introduction section to better explain transportability and included relevant references with further information.

Introduction:

“In causal inference, transportability is the extent to which effects estimated in an experimental population are correlated with effects in a target population reflecting underlying causal mechanisms^{11,12}. An *in-vitro* cellular model is ‘transportable’ to an *in-vivo* outcome phenotype if perturbation effects estimated in the *in-vitro* experiment mirror perturbation effects on the *in-vivo* outcome phenotype. Transportability requires the *in-vitro* model to effectively proxy for *in-vivo* cellular phenotypes which have downstream effects on the *in-vivo* outcome phenotype (**Figure 1A**). An *in-vitro* cellular model is an effective proxy if it has highly correlated features with pathophysiological processes of the *in-vivo* phenotype, e.g., similar biological contexts and cell-type composition. If a model is transportable, we would expect to observe that perturbations impacting the *in-vitro* cellular phenotype also impact the *in-vivo* phenotype with consistent effect sizes and directionality corresponding to the mechanistic relationships between phenotypes.”

GPAT estimates will capture a combination of both (1) effective proxy and (2) causal relationships between *in-vivo* phenotypes. To illustrate assume we have an *in-vitro* perturbation screen measuring cholesterol in hepatocytes aiming to

capture the aetiology of *in-vivo* hepatocytic cholesterol. If unsuitable experimental conditions lead to minimal correlation between *in-vitro* experimental perturbation estimates and the true *in-vivo* perturbation effects, then this will also impact any downstream GPAT analyses which combine the *in-vitro* perturbation estimates with human pLoF data. If there is minimal evidence of transportability, GPAT cannot determine whether this is because the *in-vitro* experiment was not capturing the intended phenotypes or if there is no causal relationship between the *in-vivo* cellular and outcome phenotypes.

Relatedly, the authors mentioned several times that in-vitro experiments are usually conducted in a narrow biological context while in-vivo phenotypes are in a wide biological context, and results should be interpreted carefully. Is it correct that GPAT assumes the narrow biological context proximate to the wider context? I am still a bit confused about the interpretation of "transportability" in practice. If an in-vitro model is transportable, what potential biological conclusions can be made? Could the authors clarify it further?

We hope that the introduction changes documented above on transportability have better clarified the key implications of transportability. From a data perspective: perturbation effect estimates from the *in-vitro* experiment are correlated with / predictive of perturbation effects on the *in-vivo* outcome. From a biological perspective: transportability is a consequence of underlying causal relationships and so can potentially provide insights into aetiology.

Germline genetic data captures effects of genetic differences across a wide biological context (the life-course and in all cells). In theory, GPAT could use somatic association data in specific tissues, so the assumption is more specific to use of germline genetic data in GPAT rather than GPAT itself. The discussion on biological context is relevant to interventions. For example, a therapeutic intervention may perturb a gene in a single cell-type. Such perturbation may have a different effect on a phenotype to perturbation in all cells across the life course.

2. The issue of scale mismatch between gene perturbation effects and burden-based LoF effects remains central to understanding how to interpret GPAT results. While the authors now mention a "consistency" assumption and briefly discuss it in the Discussion and Methods, this assumption is crucial to interpreting the meaning of GPAT estimates and should be brought forward earlier in the manuscript (perhaps in the Introduction or early Results).

As suggested, we have brought forward the consistency assumption to the results section.

“An additional GPAT specific assumption is that effect estimates from the *in-vitro* and *in-vivo* studies have been rescaled such that the effects reflect equivalent changes in gene function (consistency).”

Is it correct that the baseline model in simulation reflects the scenario where the consistency assumption is violated, and the estimates will be biased, but the inference will be fine?

Yes, that is correct because the directionality is consistent. In response to Reviewer 3 we have expanded the simulations to include instances where the directionality is inconsistent, and we have made further changes to the methods noted in the following point.

In particular, it would be valuable to discuss:

a. When the consistency assumption is violated, and what are the consequences? Is the non-linearity of LoF score also a source of violation?

b. The feasibility and future strategies for harmonizing scales between LoF variants and experimental perturbations. Currently, I don't have a good sense of how to harmonize these effects, especially given that the LoF effects might be non-linear.

We have expanded the methods section on consistency to discuss the consequences of violation of consistency, noting non-linearity as a potential violation. We have also expanded on discussion of how perturb-seq data could be used to harmonise effect estimates.

The impact of non-linearity of effects in gene function is unclear because of limited understanding of how commonly this occurs. There is evidence for many genes of allelic series which provide support for linear effects of gene function. For example, a common variant in a gene has a small effect on a phenotype and then a missense variant has a larger effect and a pLoF variant has the largest effect of the three.

Methods:

“An additional GPAT specific assumption is that effect estimates from the *in-vitro* and *in-vivo* studies are harmonised such that they reflect equivalent

perturbations on gene function (consistency). For example, if the effect of the *in-vitro* experimental perturbation on gene function is 2.5 SD units and the effect of *in-vivo* LoF perturbation on gene function is 1 SD units that they are scaled accordingly. Perturbations effect estimates from different data sources can be better harmonised using perturb-seq data³³ which measures the impact of perturbations on gene expression. This would enable some rescaling of experimental estimates to a 50% reduction in gene expression for more direct comparison with pLoF estimates. However, such rescaling is sensitive to the assumption of equivalence between gene expression and function and to non-linear effects of changes in function. In practice perturb-seq data is not always available for all experimental screens limiting generalisable harmonisation. If perturbation effects are directionally consistent but not perfectly harmonised, then effect estimates can be biased but causal inference is unlikely to be impacted. If perturbation effects are not directionally consistent (e.g. experimental perturbation increases gene function) then causal inference can be impacted.”

Reviewer #2 (Remarks to the Author):

Thank you to the authors for thoroughly addressing my previous comments, which has significantly improved the manuscript. I would like to recommend the revised manuscript for publication, with one minor suggestion:

The authors include reference 32 (PMID: 24995866) to summarize existing rare variant tests (e.g., burden, SKAT, ACAT). However, it should be clarified that ACAT is a P-value combination method, while the corresponding rare variant test is ACAT-V, which is not covered in reference 32. For completeness and consistency, I suggest including PMID: 32839606 alongside reference 32, as it summarizes all three tests across variant masks combining different classes of in-silico predictions.

Thank you to the reviewer for the useful feedback. We have included the additional reference as suggested.

Reviewer #3 (Remarks to the Author):

The authors have insufficiently addressed the concerns raised in my initial

review.

1.Causality versus Transportability.

The authors state: “As discussed above, we have moved away from ‘causality’ terminology to ‘transportability’ of in-vitro models which negates some of the necessity of strong assumptions.”

This response is unsatisfactory. Mendelian Randomization (MR) depends on strong assumptions to infer causality from observational data. Replacing “causality” with “transportability” does not address these fundamental concerns. Instead, it reframes the approach as an association method, which is no longer compatible with an MR framework. If the validity of transportability is uncertain, the experimental perturbations cannot serve as a valid instrument. Instruments, by definition, must be reliably associated with the exposures of interest. Without this foundation, the proposed method no longer qualifies as a valid MR analysis and would require substantial methodological revision. Accordingly, all words related to MR and instrumental variables should be removed.

Thank you to the reviewer for the useful feedback.

Previously we did not provide sufficient detail on transportability. We have rewritten the introduction section to better explain transportability and included relevant references with further information. Hopefully this rewritten introduction better explains transportability and clarifies that transportability requires causal inference. There is a substantial literature on ‘transportability’ in the causal inference field by Judea Pearl and others relating to transporting effects from an experimental population to a different context. For example, can effect estimates from a randomised trial in Los Angeles be extrapolated to New York?

An *in-vitro* cellular phenotype cannot technically have causal effects on an *in-vivo* phenotype complicating the terminology required for GPAT (in comparison to MR). However, clearly experimental perturbation estimates can still capture effects relating to underlying causal mechanisms and so (depending on transportability) enable inference about the aetiology of *in-vivo* phenotypes.

Introduction:

“In causal inference, transportability is the extent to which effects estimated in one experimental population correlate with the effects in another population, reflecting stable underlying causal mechanisms^{11,12}. An *in-vitro* cellular model is ‘transportable’ to an *in-vivo* outcome phenotype if perturbation effects estimated in the *in-vitro* experiment mirror perturbation effects on the *in-vivo* outcome phenotype. Transportability requires the *in-vitro* model to effectively proxy for *in-vivo* cellular phenotypes which have downstream effects on the *in-vivo* outcome

phenotype (**Figure 1A**). An *in-vitro* cellular model is an effective proxy if it has highly correlated features with pathophysiological processes of the *in-vivo* phenotype, e.g., similar biological contexts and cell-type composition. If a model is transportable, we would expect to observe that perturbations impacting the *in-vitro* cellular phenotype also impact the *in-vivo* phenotype with consistent effect sizes and directionality corresponding to the mechanistic relationships between phenotypes.”

2. The newly added simulations are insufficient. They only demonstrate directional consistency, which does not adequately capture the real-world complexity. In practice, heterogeneity between experimental and germline LoF perturbations can lead to effects in opposing directions. These discrepancies pose a serious challenge to the proposed framework and must be more thoroughly investigated.

In practice, directional consistency between effects of pLoF variants and experimental perturbations is likely to be the norm rather than the exception. Indeed, in-silico predictions are generally very accurate for LoF variants and experimental screens can evaluate if perturbations are impacting the expression of the gene as expected. If missense variants were included then directionality would definitely be a problem but as discussed previously, we advise to include only pLoF variants.

However, we acknowledge that there are likely to be some instances of misannotation leading to directional inconsistency. We have therefore expanded the simulations to include an additional fifth model. This model is identical to the baseline model but pLoF burden estimates for 20% of genes have their direction of effect reversed (i.e. actually reflect gain of function; inconsistent directionality). As expected, this leads to additional noise in inference and bias in the overall estimate with the impact of bias dependent on the degree of misannotation. Please find relevant excerpts below.

Methods:

- Directional inconsistency model:
 - Baseline model with 20% of V_j^{LoF} values having their direction flipped to instead be gain-of-function.
 - This results in some experimental perturbations and LoF variants having inconsistent directionalities.

A key element of the baseline model is that differences in perturbation magnitude between experimental (mean = 1.25 SD units) and LoF burden (mean = 0.75 SD units) perturbations were not controlled for. GPAT IVW

estimates were biased downwards (expected mean value = 1, observed = 0.56; 95% CI 0.54, 0.58) similar to the ratio of perturbation effects (0.6). **The directional inconsistency model showed further attenuations towards the null due to misannotation of pLoF burden estimates for 20% of genes which actually reflected effects of gain of function (observed mean estimate = 0.36; 95% CI 0.34, 0.38).** These results illustrate that GPAT estimates are sensitive to directional consistency and harmonization in magnitudes between the two perturbation sources (**Supplementary Table 9**).

Plot (Supplementary Figure 10):

3. Lack of Comparison with alternatives. The authors claim to have addressed comment #4, but their response is inadequate. Only discussion is not enough. A rigorous comparison with existing methods using empirical data or simulations is essential to demonstrate the advantages of the proposed approach. Without such evidence, the method appears to be just another association-based method, at best an integrative association across data sources, making it difficult to evaluate its validity, robustness, or practical utility relative to current alternatives.

As noted in response to the first comment above, GPAT is a causal inference approach attempting to evaluate model transportability. We hope that the revised introduction better clarifies this and illustrates the advantages over 'association' methods such as gene-set enrichment.

We have acknowledged in the discussion that GPAT is not without limitations and so is best used in a triangulation of evidence framework alongside other complementary approaches. We have included comparisons with enrichment in the manuscript for hypothesis-driven GPAT analyses and illustrated an example where the methods lead to different conclusions (chondrocytes and height) with the GPAT inference more robust. Comparisons with other methods such as differential expression analysis are beyond the scope of this manuscript as they require completely different datasets and models.

Discussion:

“GPAT enables hypothesis-free and systematic evaluation of the transportability of *in-vitro* cellular models to *in-vivo* organismal phenotypes. The method is sensitive to certain assumptions and so we recommend its application within a triangulation of evidence framework³² alongside complementary *in-vitro* methods and genomic approaches (e.g., differential expression analysis). The potential applications of GPAT are likely to accelerate over the next decade, driven by increased functional genomics data, larger datasets of sequencing data and improved variant predictions.”

Reviewer #4 (Remarks to the Author):

The authors have made substantial changes to the manuscript; including terminology, explanations, limitations, and new analyses. The manuscript is improved, and I do not have further comments.

Thank you to the reviewer for the useful feedback.

REVIEWERS' COMMENTS

Reviewer #1 (Remarks to the Author):

The authors have addressed the concerns raised in my previous round of review. I am satisfied with the paper. Thank you!

Thank you for the helpful feedback on the manuscript.

Reviewer #3 (Remarks to the Author):

The authors have addressed most of the previous concerns, and the revised manuscript shows substantial improvement in clarity and rigor. The assumptions and limitations are now clearly articulated, and the revised introduction and the added simulations and explanations strengthen the overall presentation. A minor suggestion: The discussion needs to appropriately acknowledge the weaknesses and the sensitivity to assumptions.

Thank you for the helpful feedback on the manuscript. We have modified the final paragraph to more prominently acknowledge the limitations of the proposed framework.

Before:

GPAT enables hypothesis-free and systematic evaluation of the transportability of *in-vitro* cellular models to *in-vivo* organismal phenotypes. The method is sensitive to certain assumptions and so we recommend its application within a triangulation of evidence framework³³ alongside complementary *in-vitro* methods and genomic approaches (e.g., differential expression analysis). The potential applications of GPAT are likely to accelerate over the next decade, driven by increased functional genomics data, larger datasets of sequencing data and improved variant predictions.

After:

GPAT enables hypothesis-free and systematic evaluation of the transportability of *in-vitro* cellular models to *in-vivo* organismal phenotypes. The method has limitations and is sensitive to certain assumptions, notably germline human genetic data capture broad organism-level life course effects which may differ to therapeutic effects in a narrower context. We therefore recommend its application within a triangulation of evidence framework³³ alongside complementary *in-vitro* methods and genomic approaches (e.g., differential expression analysis). The potential applications of GPAT are likely to accelerate over the next decade, driven by increased functional genomics data, larger datasets of sequencing data and improved variant predictions.